# Design of Novel Laser Crosslink Systems Using Nanosatellites in Formation Flying: The VISION

**Geuk-Nam Kim** [1], **Sang-Young Park** [1,*], **Sehyun Seong** [2], **Jae-Young Choi** [3], **Sang-Kook Han** [3], **Young-Eon Kim** [1], **Suyong Choi** [1], **Joohee Lee** [1], **Sungmoon Lee** [1], **Han-Gyeol Ryu** [1] and **Seonghui Kim** [2,4]

1 Astrodynamics and Control Laboratory, Department of Astronomy, Yonsei University, Seoul 03722, Korea; south1003@yonsei.ac.kr (G.-N.K.); 0-eon@yonsei.ac.kr (Y.-E.K.); sycelot@yonsei.ac.kr (S.C.); schercho528@yonsei.ac.kr (J.L.); leesungmoon@yonsei.ac.kr (S.L.); morimn21@gmail.com (H.-G.R.)
2 Telepix Co., Ltd., Busan 48058, Korea; shseong@telepix.net (S.S.); barlow@telepix.net (S.K.)
3 Broadband Transmission Network Laboratory, School of Electrical and Electronic Engineering, Yonsei University, Seoul 03722, Korea; chql@yonsei.ac.kr (J.-Y.C.); skhan@yonsei.ac.kr (S.-K.H.)
4 Korea Aerospace Research Institute, Daejeon 34133, Korea
* Correspondence: spark624@yonsei.ac.kr

**Abstract:** With growth in data volume from space missions, interest in laser communications has increased, owing to their importance for high-speed data transfer in the commercial and defense fields, spaceborne remote sensing, and surveillance. Here, we propose a novel system for space-to-space laser communication, a very high-speed inter-satellite link system using an infrared optical terminal and nanosatellite (VISION), which is aimed at establishing and validating miniaturized laser crosslink systems and several space technologies using two 6U nanosatellites in formation flying. An optical link budget analysis is conducted to derive the signal-to-noise ratio requirements and allocate the system budget; the optical link margin should be greater than 10 dB to guarantee communication with practical limitations. The payload is a laser transceiver with a deployable space telescope to enhance the gain of the beam transmission and reception. Nanosatellites, including precise formation flying GNC systems, are designed and analyzed. The attitude control system ensures pointing and tracking errors within tens of arcsec, and they are equipped with a propulsion system that can change the inter-satellite distance rapidly and accurately. This novel concept of laser crosslink systems is expected to make a significant contribution to the future design and construction of high-speed space-to-space networks.

**Keywords:** laser crosslink; PAT (pointing, acquisition, and tracking); nanosatellite; formation flying

## 1. Introduction

Laser communication is a promising method for dealing with the recent growth in data volume from spaceborne platforms, achieving a super-high data rate that is faster than 1 Gbps. Laser communication systems enhance the size, weight, and power (SWaP) efficiency compared to traditional radio frequency (RF) systems at low cost [1]. With a wide spectral range and narrow beam feature, this system improves link security, reducing the potential risk from mutual interference, jamming, and signal interception from others. In addition, there are no regulatory constraints on licensing frequency bands, which is helpful in establishing a low Earth orbit (LEO) mega-constellation. The applications of laser communication include commercial and defense operations as well as high-speed data relay in remote sensing or surveillance systems [2]. Fundamental technologies for space-to-ground laser communication systems have been implemented and operated on-orbit, such as GOLD, LUCE, and LLCD [3]. By utilizing nanosatellite platforms, key technologies for space-to-space, termed crosslink or inter-satellite link (ISL), can be validated on-orbit at a low development cost. CubeSat Laser Infrared CrosslinK (CLICK-B/C) is a technology demonstration mission that uses two nanosatellites for a laser crosslink in the range of

25–500 km [4]. As a part of the NASA Optical Communication and Sensor Demonstration (OCSD) program, AeroCube-7B/C demonstrated a precise pointing system with miniaturized actuators and sensors for laser crosslinks [5]. Furthermore, the laser interconnect and networking communications system (LINCS)-A/B was developed to demonstrate laser communication technologies with a data rate of 5 Gbps at an inter-satellite distance of 2000 km [6]. Inter-satellite laser communication requires precise formation flying technologies, such as relative navigation and pointing maneuvers. Canadian Advanced Nanospace eXperiment-4&5 (CANX-4&5), developed by UTIAS/SFL, demonstrated autonomous formation flying technologies, including relative navigation and positioning maneuvers [7]. Gomspace eXperiment-4A&B (GomX-4A&B), led by GomSpace, demonstrated RF crosslink technology and implemented orbit maneuvers by changing the inter-satellite distances using a cold-gas propulsion system [8]. Yonsei University has developed CubeSat Astronomy by NASA and Yonsei using Virtual ALignment-eXperiment/Coronagraph (CANYVAL-X/C) to study the core technologies of a virtual space telescope based on autonomous formation flying [9].

To implement the laser crosslink and demonstrate several space technologies, we propose the very high-speed inter-satellite link system using infrared optical terminal and nanosatellite (VISION) mission. This study is aimed at establishing and validating high-speed and miniaturized laser crosslink systems using two 6U nanosatellites in formation flying, termed as Altair and Vega. The final goal is to achieve a data rate of 1 Gbps with a coded bit error rate (BER) of less than $1 \times 10^{-9}$ at an inter-satellite distance of 1000 km. Figure 1 presents a conceptual illustration of the VISION mission. The total system has half the mass and size of the LINCS system [6], with similar performances for the laser crosslink. The mission payload of the laser communication terminal (LCT) is equipped with deployable segmented front-end optics (FEO), which enhances the gain of beam transmission and reception relevant to a large aperture. The proposed deployable optics can be applied to high-resolution Earth observation payloads with miniaturization and mass reduction. For a laser crosslink, a pointing, acquisition, and tracking (PAT) system is required. The proposed systems are applied to a monostatic architecture that shares the beam path for communication and PAT with a single aperture. This scheme can mitigate a steady-state pointing error with closed-loop feedback based on a fast-steering mechanism (FSM), reducing the residual line-of-sight (LOS) jitter to less than 1 μrad. The LCT is integrated with a 6U nanosatellite bus, which has a precise pointing system for LOS alignment during laser crosslinking. To generate an accurate target LOS vector, a relative navigation algorithm using both GPS L1 and GPS L2 signals has been designed to mitigate ionospheric delay effects for a long baseline. The two satellites are equipped with a propulsion system for rapid and accurate orbit maneuvers to adjust inter-satellite distances.

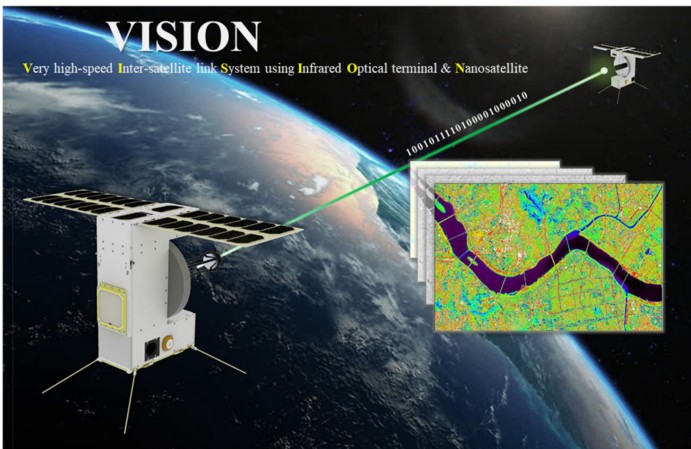

**Figure 1.** Conceptual illustration of the VISION mission. The proposed system can transfer large-sized image data and will be applied to construct a high-speed data relay system in future.

The contributions of this study are as follows. First, a novel system architecture for future laser crosslinks is proposed. In the last ten years, the key technologies of laser communication systems have been developed and validated by on-orbit missions, adopting large-sized satellite platforms with additional beam-pointing and tracking systems such as gimbal mechanisms [10]. The proposed systems share beam paths for communication and PAT with a single aperture. As the PAT is only assisted by the nanosatellite bus, the systems require precise feedback control combined with bus attitude and FSM operations. This challenging architecture can significantly reduce the steady-state errors of beam paths between communication and tracking, not only by reducing the system size but also by enhancing communication performance.

Second, the design processes of a practical laser crosslink mission and related systems are outlined. The system budgets were allocated from the optical link budget design and analysis, including on-orbit noise estimations such as incident sunlight and thermal effects on the detectors. The handover process of the crosslink, applying a monostatic architecture, was designed for link access and maintenance. In addition, considering new technologies for space optics, such as FEO, it is important to evaluate the communication performance based on the optical link budget. The on-orbit operation scenario includes laser crosslink tests with various inter-satellite distances [11]. Unlike other systems utilizing nanosatellite platforms, the proposed systems include a propulsion system for rapidly and accurately changing the distances. Additionally, considering the long baseline, a relative navigation algorithm handling ionospheric delay was designed and analyzed. We adopted a standardized and commercial-off-the-shelf (COTS)-based platform, which would contribute to the agile construction and maintenance of LEO mega-constellations for global networks in the future.

Finally, fundamental technologies for advanced space-optical systems are proposed. In particular, a deployable space telescope (DST) is applied to the FEO to improve the optical link performance with a large aperture, which is up to 10 times wider than that of other systems. The optics are composed of three segmented reflectors for the primary mirror and a boom mechanism for the secondary mirror. This deployable optics technology can be applied to Earth observation missions for super-high-resolution images using Cube-/nanosatellite platforms, significantly reducing their size and mass [12]. Furthermore, miniaturized back-end optics were designed utilizing COTS-based optics and FSM, enabling rapid development. The integrated front-end and back-end optics design was optimized to increase the received signal power and enhance the SNR (signal-to-noise ratio) margin of the optical link channel.

The remainder of this paper is organized as follows. Section 2 describes the laser crosslink mission in detail, including the requirements and concept of operations. Section 3 covers the nanosatellite system design specifications for the laser crosslink, as well as the optical link budget analysis for each scenario. Sections 4 and 5 describe the design and analysis of payload and bus systems, respectively. Finally, Section 6 summarizes the study and presents concluding remarks.

## 2. Laser Crosslink Mission

### 2.1. Mission Statement

The VISION mission is aimed at demonstrating novel laser crosslink systems using two nanosatellites, achieving a data rate of 1 Gbps at 1000 km apart. To establish the crosslink, the optical axes of each nanosatellite are aligned precisely, reducing the residual jitter to a LOS smaller than 1 µrad to ensure the optical link performance. Table 1 lists the top-level mission requirements (TMR) and constraints for the system design. The mission lifetime should be longer than one year, and the systems are enveloped in the standard 6U nanosatellite.

**Table 1.** Top-level project requirements and constraints.

| Identification | Description |
|---|---|
| TMR.001 | Mission lifetime shall be longer than 12 months. |
| TMR.002 | The laser crosslink shall be established for the inter-satellite distance up to 1000 km. |
| TMR.003 | The data capacity shall be faster than 1 Gbps at $1 \times 10^{-3}$ of uncoded BER and $1 \times 10^{-9}$ of coded BER. |
| TMR.004 | A precision of residual line-of-sight jitters shall be smaller than 1 µrad while the laser crosslink is established. |
| TMR.005 | Entire laser crosslink systems shall be contained in the 6U nanosatellite platforms |

*2.2. Orbit Scenarios*

The mission lifetime is composed of three phases: the launch and early orbit phase (LEOP); the drift recovery and station-keeping phase (DRSKP); and the normal operation phase (NOP). These phases comprise several modes of system check-out, telecommunication, hardware commissioning, and maneuvers. The concept of operations (ConOps) of the nanosatellites is presented in Figure 2. After being ejected in orbit, they drift several thousand kilometers away and operate independently. Through orbit maneuvers in the DRSKP, the inter-satellite distance is reduced. During the NOP, they sequentially adjusted the inter-satellite distance from 50 km to 1000 km and conducted laser crosslink tests. Figure 3 shows the various inter-satellite distances over the mission lifetime. The details of ConOps are as follows:

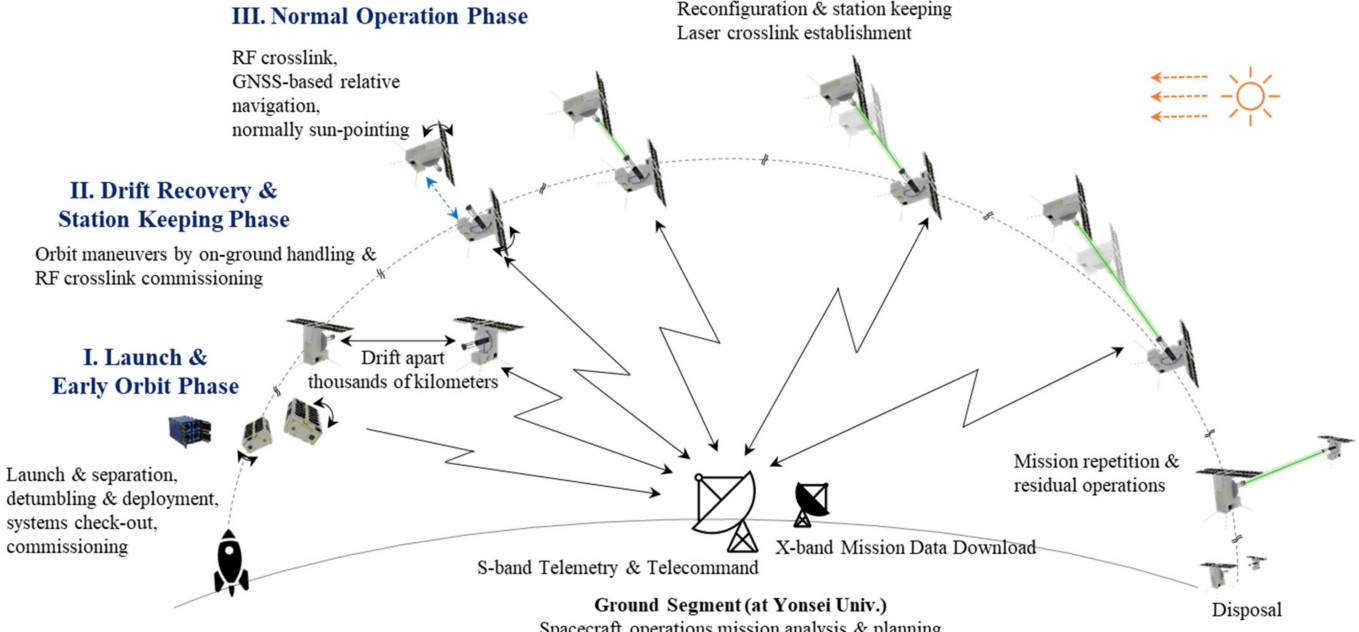

**Figure 2.** Concept of operations for the VISION mission. After the early orbit operations, the two CubeSats adjust their distances. For each specific distance, they maintain over 10 days to conduct the laser crosslink tests.

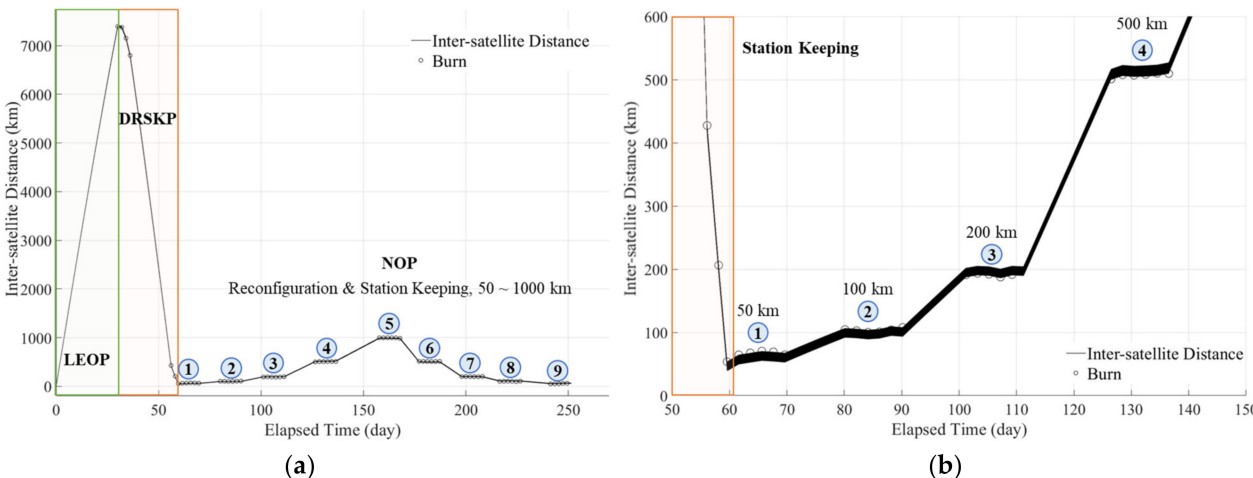

**Figure 3.** Profiles of inter-satellite distance with orbit maneuver scenarios using multiple burns. Blue circles denote a repeat count of the reconfiguration and station keeping maneuvers: (**a**) entire phases from launch and separation; (**b**) normal operation phase with reconfiguration and station-keeping maneuvers for adjusting inter-satellite distances.

### 2.2.1. Phase I—Launch and Early Orbit Phase (LEOP)

Following separation from the launch vehicle, Altair and Vega deployed a UHF antenna and started transmitting the beacon signals. Subsequently, they stabilized their initial spins using three-axis reaction wheels (RWs). When the detumbling operation ended, they deployed solar panels to maximize battery charging. Commissioning operations were performed to calibrate the components, including RF devices and attitude sensors. Before the orbit maneuvers, the propulsion system was commissioned to evaluate and calibrate the firing logic conditions, enabling heaters on the propellant tank and each nozzle. In addition, the optical properties of the payload were calibrated by measuring the on-orbit background noise of the sensors for tracking and communication. During the LEOP, the inter-satellite distance gradually increased to over a few thousand kilometers owing to the perturbation forces.

### 2.2.2. Phase II—Drift Recovery and Station-Keeping Phase (DRSKP)

In the DRSKP, the thrusters were fired towards the desired directions, as determined by the flight dynamics system (FDS) and mission planning system (MPS), until the distance between nanosatellites was reduced to approximately 50 km. While decreasing the distances, commissioning operations for the RF crosslink and laser crosslink were performed to verify the on-orbit availability of the crosslinks and the relative navigation. Finally, station-keeping maneuvers were performed to satisfy the initial relative distance maintenance.

### 2.2.3. Phase III—Normal Operation Phase (NOP)

In the NOP, fine relative navigation based on the differential GPS (DGPS) algorithm was implemented for satellites to seek one another [13]. Moreover, to evaluate the optical link performance, orbit maneuvers were conducted for 50, 100, 200, 500, and 1000 km baselines: for several laser crosslink tests, the distance between satellites was kept within 10% of each baseline over 10 days. When the LOS was precisely aligned, the sensors on the LCT detected beams from each other. Finally, the PAT system operated to maintain the spot position by utilizing the FSM within the active area of the detectors. While they carry out the laser crosslink, the solar panels orienting the sun can mitigate a solar direct noise incident on the LCT detectors. In addition, the GNSS antenna and star tracker's aperture stare the zenith to enhance visibility, improving the navigation performance. The mission operations were repeated by varying the baseline throughout the mission lifetime.

### 2.2.4. Operation Modes

Figure 4 shows that each phase consists of several modes, and it describes their flow. After the end of detumbling in the LEOP, both nanosatellites enter the standby mode by the ground telecommand. The standby, communication, and safe modes are autonomously exchanged and activated by monitoring the systems' statuses, such as battery capacity, temperature, telecommand schedule, etc. In standby mode, they are ready for receiving telecommands to change the mode and synchronize the time with ground station, periodically monitoring the status and schedule. The telemetry and mission data are downloaded during the communication mode. The safe mode, having the highest priority, handles the systems contingencies, such as solar panels sun-pointing to charge the batteries, detumbling to stabilize body spins, and autonomous execution of back-up operations to mitigate risks induced by a communication fail. The commissioning, maneuver, and mission modes can be entered by ground telecommands. In the NOP, the commissioning is relevant to formation flying for the PAT. Finally, the mission mode is defined as the entire sequence for the laser crosslink, and is described in the following subsection. When each mode operation is completed, they get back to the standby mode automatically.

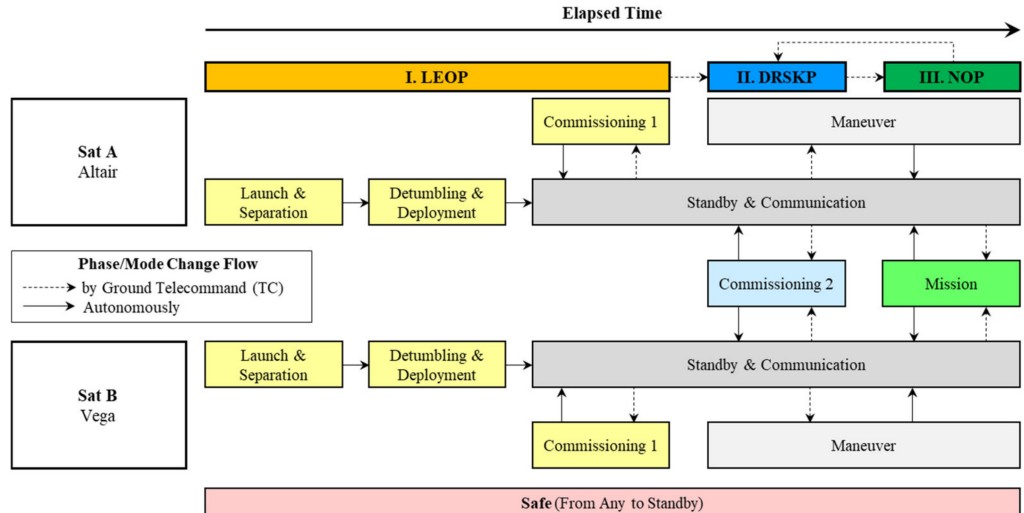

**Figure 4.** Operations mode flows diagram.

### 2.3. Pointing, Acquisition, and Tracking (PAT) Sequence

To accomplish the laser crosslink, the LOS between each satellite is aligned precisely using body pointing and the FSM, which is termed the PAT sequence. The PAT sequence is classified into three stages: (1) bus initialization stage (BIS), (2) coarse PAT stage (CPS), and (3) fine PAT stage (FPS). Specifically, for the CPS and FPS, two types of beam divergence angles and sensors are used as beam detectors; one is the SWIR CAM (short-wave infrared camera) detector, termed the CAM, and the other is the Quadrant Cell (QC) detector. Figure 5 shows the PAT sequence in nanosatellite orientations with various beam divergence angles.

Before the bus initialization operation, the nanosatellites locate each other by pointing to a target predicted by the mission-planning system of the ground segment. When they establish the RF crosslink, relative navigation is started to calculate the LOS vectors. The star tracker on each satellite is used to acquire its spatial orientation. The attitude maneuver using 3-axis RWs aligns the LOS with each other and compensates errors induced by orbital motions. When the estimated LOS error is smaller than the full field of view of the CAM or the field of uncertainty (FOU), they start to transmit a broad beam. The beam spot projected on the CAM is biased due to control errors and mechanical misalignment: the bias is defined as the AOA (angle of arrival). Applying the CAM feedback and attitude maneuver, each satellite sequentially corrects the AOA until the beam spot on both sides remains within a threshold of the FFOV of the QC. However, the jitter induced by the

platform remains. The FSM is activated to eliminate jitter. When entering the FPS, the beam divergence angle is narrow, and the QC is used for the FSM feedback with high-frequency measurements. Within the tip-tilt angle of the FSM, the LOS errors from the attitude maneuver and jitter residuals can be rejected, enhancing the optical link performance.

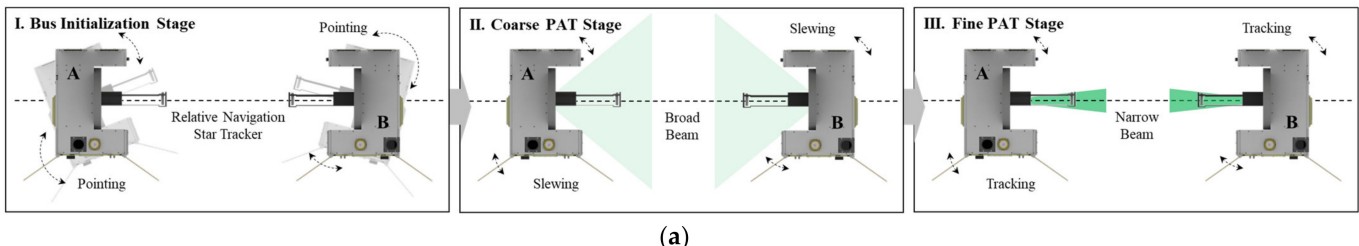

(a)

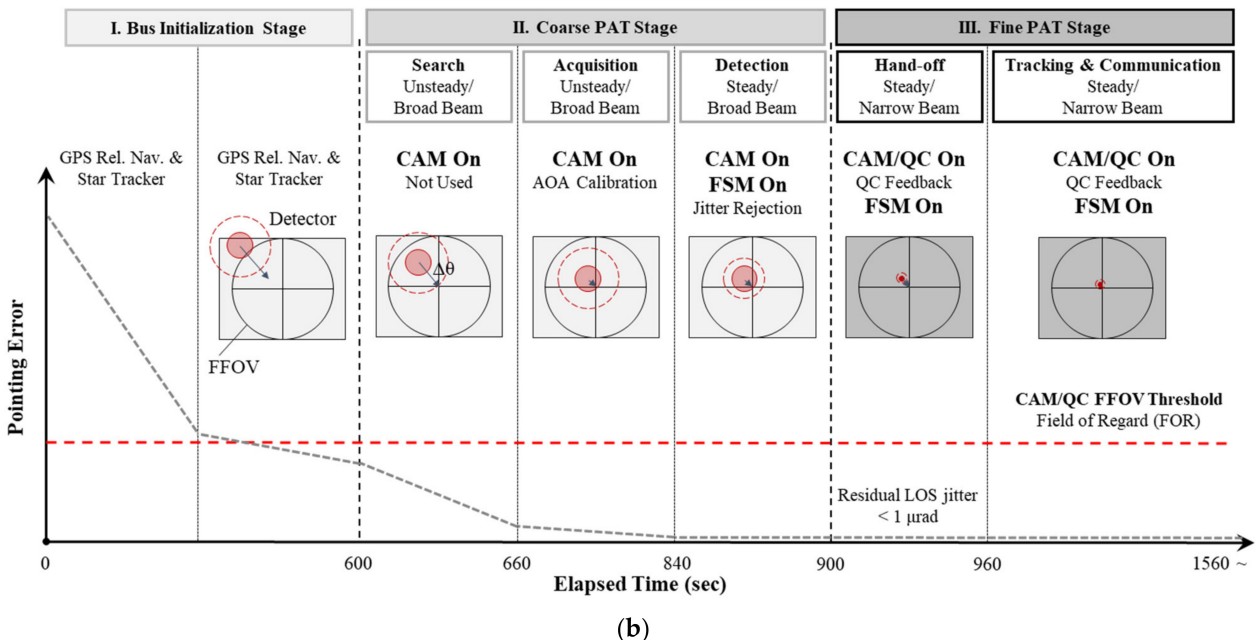

(b)

**Figure 5.** Pointing, acquisition, and tracking (PAT) sequence for laser crosslink: (**a**) nanosatellite orientations and operations, depicting the beam divergence angle; (**b**) beam spot on tracking sensor and pointing error profiles over substages. After the Acquisition substage is ended, the FSM is available to reject jitters.

Table 2 summarizes the details of the PAT sequence substages. The CPS should end within 5 min, and the FPS should be maintained for over 10 min. During the search operation, the CAM is used only to check whether the beam is projected within the active area of the detector. For the acquisition and detection operations, CAM provides the AOA to be corrected by attitude maneuvers. During detection, the FSM is used to reduce jitter. The pointing error over the CPS should be smaller than 1200 μrad and 400 μrad for the bias and standard deviation errors, respectively. For FPS, the precision of the LOS jitter should be less than 1 μrad. During tracking and communications, the CAM is operated as a backup detector to prevent the nanosatellites from missing each other. Furthermore, the RF crosslink is always enabled to execute relative navigations. Considering the bus and LCT operations, the optical link budget cases are classified into five cases for either PAT or communication.

**Table 2.** PAT sequence descriptions.

| Contents | Coarse PAT Stage (CPS) | | | Fine PAT Stage (FPS) | |
|---|---|---|---|---|---|
| **Substage** | **Search** | **Acquisition** | **Detection** | **Hand-Off** | **Tracking and Communication** |
| **Control** Duration [sec] | ~60 | ~180 | ~60 | ~60 | >600 |
| Pointing Error ($\mu$, $\sigma$) [(1)] [$\mu$rad] | | <(1200, 400) | | | <(30, 1) |
| **Bus Operation** Attitude Control | Slewing | AOA Correction | Slewing | | Slewing |
| RF Crosslink | | Enabled/ Relative Navigation | | | Enabled/ Relative Navigation |
| **LCT Operation** Beam | | Broad/Unsteady | Broad/Steady | | Narrow/Steady |
| Actuator | | | FSM On | | FSM On |
| Detector | CAM On/ Unavailable | | CAM On/ Available | | QC (CAM [(2)]) On/ Available |
| **Optical Link Budget Case** | PAT#1 | PAT#2 | PAT#3 | PAT#4 | COM#1 |

[(1)] $\mu$: bias errors, $\sigma$: standard deviations errors. [(2)] Only QC is used for feedback, CAM is available to prevent nanosatellites from missing each other.

## 3. Concept of the Laser Crosslink Systems

### 3.1. Systems Architecture

Figure 6 presents the system architecture of the integrated laser communication payload and nanosatellite bus. The diagram describes the electrical interfaces, including the power supply and data communication. The system should achieve a data rate of 1 Gbps at an inter-satellite distance of 1000 km, securing an SNR margin higher than 10 dB. The size should then be fitted to the 6U nanosatellite standard.

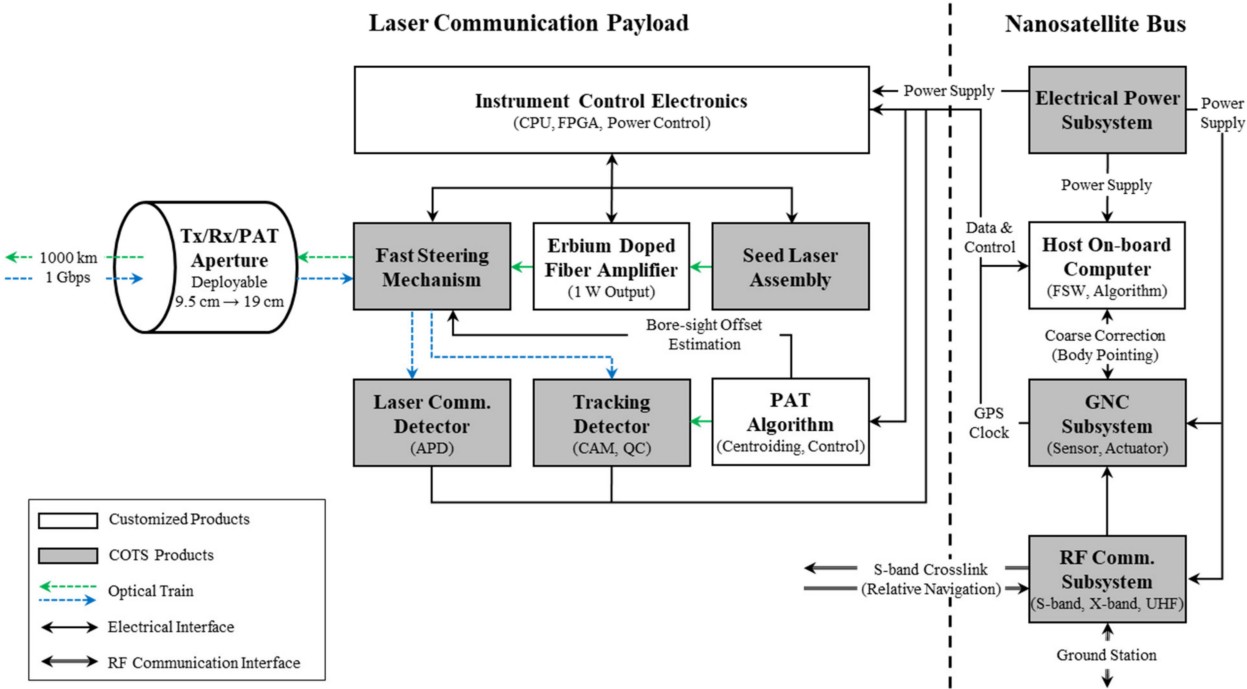

**Figure 6.** Diagram of the integrated laser communication payload and nanosatellite bus systems architecture.

The optical train of the payload is a monostatic architecture that shares the aperture for laser communication and PAT. Unlike other laser communication systems with an extra beam-tracking system, the proposed system mitigates steady-state errors for feedback control during the PAT sequence. The aperture has a deployable configuration to enhance the optical power and minimize the stowed size during the launch phase. Moreover, an erbium-doped fiber amplifier (EDFA) pumps a laser generated by the seed laser assembly (SLA) up to 1 W for long-range data transmission. The ratio of laser powers for communication and PAT is 99:1. The instrument control electronics (ICE) of the payload manages the payload electronics, including the regulation and distribution of electrical power provided by the electrical power subsystem (EPS) of the bus.

The OBC (on-board computer) handles the entire system using FSW (flight software). The guidance, navigation, and control (GNC) subsystem assists the PAT with precise attitude maneuvers. In addition, it provides GPS clock signals for the payloads to synchronize with each other. The RF communication subsystem performs S-band crosslink sharing of the GPS raw data to establish the relative navigation in a long baseline.

### 3.2. Optical Link Budget

There are several power penalty sources: pointing loss, optics loss, fiber coupling loss, electronics degradation, and sensor noise. The SNR margin should be higher than 10 dB to guarantee the optical link performance of the VISION systems. The optical link budget analysis process refers to the case studies of CLICK-B/C [14].

### 3.2.1. Pointing Error Budget

Pointing loss due to pointing errors is dominant in the optical link performance degradation of laser crosslink missions. The pointing error budget is classified into point-ahead and tracking terms, as shown in Figure 7. The point-ahead errors contain body pointing and mechanical misalignment, which can be corrected using CPS. Tracking errors are related to the signal noise of the detectors and jitter residuals, which are reduced over the FPS.

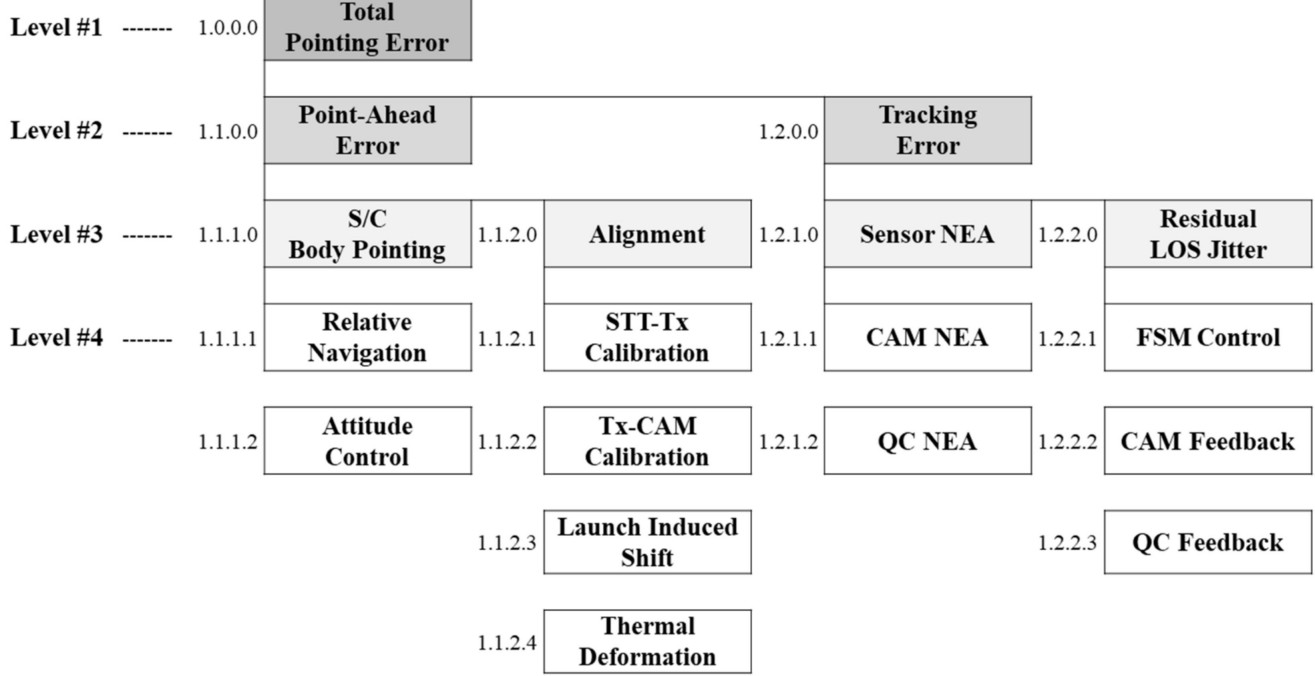

**Figure 7.** Diagram of the pointing error budget structure.

The pointing errors for each PAT sequence at inter-satellite distances of 50 km and 1000 km are summarized in Table 3. For PAT#1 and PAT#2, only attitude maneuvers were conducted to roughly correct the AOA. During PAT#3, the bias of point-ahead errors was mostly eliminated. Finally, at PAT#4 and COM#1, the systems can achieve a total pointing error smaller than 1 μrad with the FSM control.

**Table 3.** Pointing errors elements and values for each PAT sequence.

| Pointing Error | | PAT#1 (μ, σ) [μrad] | | PAT#2 (μ, σ) [μrad] | | PAT#3 (μ, σ) [μrad] | | PAT#4 and COM#1 (μ, σ) [μrad] | |
|---|---|---|---|---|---|---|---|---|---|
| **Elements** | | **50 km** | **1000 km** | **50 km** | **1000 km** | **50 km** | **1000 km** | **50 km** | **1000 km** |
| **Point-Ahead** | Body Pointing | (386.76, 171.67) | (310.61, 109.99) | (384.79, 171.67) | (307.86, 109.99) | (0.33, 2.42) | (6.67, 2.43) | | |
| | Alignment | (223.21, 173.40) | (223.21, 173.40) | (223.21, 173.37) | (223.21, 173.37) | (0.00, 173.21) | (0.00, 173.21) | | |
| **Tracking** | Sensor Noise | | | (0.00, 0.40) | (0.00, 8.00) | (0.00, 0.40) | (0.00, 8.00) | (0.00, 0.10) | (0.00, 0.60) |
| | Residual Jitter | | | (0.00, 4.00) | (0.00, 4.00) | (0.10, 4.06) | (0.10, 4.06) | (0.10, 0.76) | (0.10, 0.76) |
| **Total** | | (609.96, 244.00) | (533.82, 205.34) | (608.00, 244.02) | (531.06, 205.51) | (0.43, 173.27) | (6.77, 173.45) | (0.10, 0.77) | (0.10, 0.97) |

### 3.2.2. Signal-to-Noise Ratio Margin

When the beam is propagated in free space, the signal power is reduced inversely proportional to the square of the inter-satellite distance, as shown in Figure 8. Given the aforementioned pointing errors, the received signal power on the laser communication detector, which is the avalanche photodiode detector (APD), is not sufficient for PAT and communication. The transmitted and received laser powers on each satellite are significantly enhanced by deployable optics. Moreover, the signal must be sufficiently detectable from the background noise of each detector. For feasible on-orbit communication with novel optics, the SNR margin must exceed 10 dB within the maximum range.

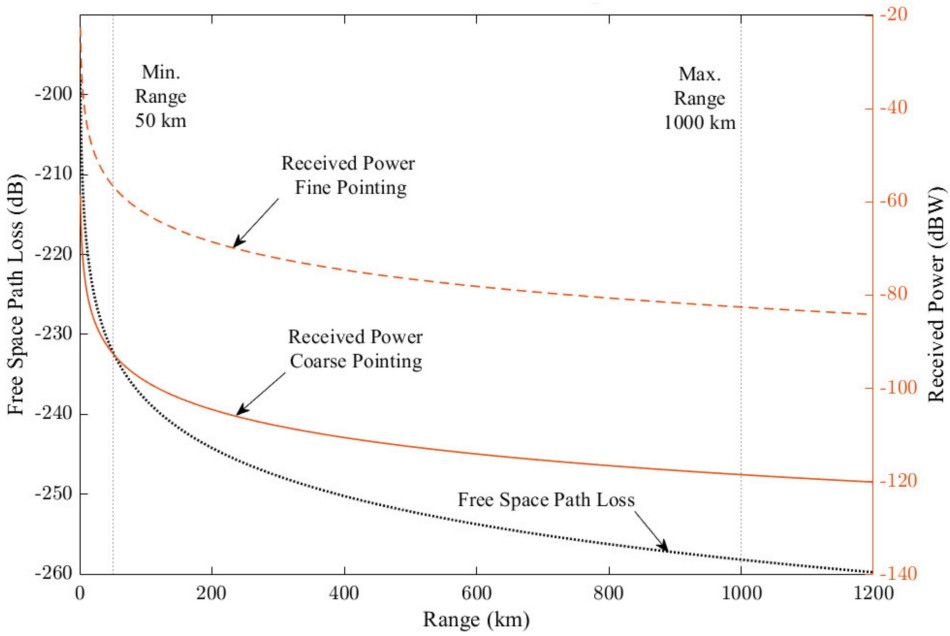

**Figure 8.** Free space path loss and received power for the laser communication detector over various inter-satellite distances.

Table 4 presents the optical link budget analysis results for the maximum range. The transmitted power is 1 W and split for the PAT and communication. For the PAT cases, the SNR margin is estimated for the CAM and QC detectors, following the pointing stages. Communication modulation is applied to on-off keying (OOK), which is primarily utilized in spaceborne laser communication systems. The required SNR was calculated according to a target BER of $1 \times 10^{-3}$. Although the beam power for each PAT sequence is weak compared with the communication, the SNR margin meets the system requirements, achieving approximately 11 dB for the CPS and 22 dB for the FPS. The narrow beam divergence angle for PAT#4 and COM#1 significantly contributed to increasing the received power and enhancing the optical link performance.

**Table 4.** Optical link budget and SNR margin for each PAT sequence in 1000 km of inter-satellite distance: (**a**) PAT#1 and PAT#2, (**b**) PAT#3; (**c**) PAT#4; (**d**) COM#1.

| (a) PAT#1 and PAT#2—Search and Acquisition | | | | |
|---|---|---|---|---|
| **Elements** | **Budget** | **Available** | **Unit** | **Remarks** |
| Tx Power | >−3 | 0.00 | dBW | 1 W output |
| Tx Gain | >56 | 56.98 | | Beam divergence, 8011.4 μrad |
| Pointing Loss | >−5 | −0.68 | | Pointing error at Table 3 |
| Tx Optics Loss | >−7 | −2.00 | dB | Front-end, back-end optics |
| Path Loss | <−259 | −258.18 | | 1000 km apart |
| Rx Gain | >111 | 111.71 | | Clear aperture diameter, 19 cm |
| Rx Optics Loss | >−7 | −7.00 | | Max. tracking sensor (CAM) |
| Rx Power | >−125 | −119.17 | dBW | Beam split, 99:1 |
| **SNR Margin** | >10 | 11.49 | dB | - |
| (b) PAT#3—Detection | | | | |
| **Elements** | **Budget** | **Available** | **Unit** | **Remarks** |
| Tx Power | >−3 | 0.00 | dBW | 1 W output |
| Tx Gain | >56 | 56.98 | | Beam divergence, 8011.4 μrad |
| Pointing Loss | >−5 | −0.19 | | Pointing error at Table 3 |
| Tx Optics Loss | >−7 | −2.00 | dB | Front-end, back-end optics |
| Path Loss | <−259 | −258.18 | | Max. range, 1000 km |
| Rx Gain | >111 | 111.71 | | Clear aperture diameter, 19 cm |
| Rx Optics Loss | >−7 | −7.00 | | Max. tracking sensor (CAM) |
| Rx Power | >−125 | −118.69 | dBW | Beam split, 99:1 |
| **SNR Margin** | >10 | 11.81 | dB | - |
| (c) PAT#4—Hand-off and Tracking | | | | |
| **Elements** | **Budget** | **Available** | **Unit** | **Remarks** |
| Tx Power | >−3 | 0.00 | dBW | 1 W output |
| Tx Gain | >56 | 92.92 | | Beam divergence, 127.8 μrad |
| Pointing Loss | >−5 | −0.03 | | Pointing error at Table 3 |
| Tx Optics Loss | >−7 | −2.00 | dB | Front-end, back-end optics |
| Path Loss | <−259 | −258.18 | | Max. range, 1000 km |
| Rx Gain | >111 | 111.71 | | Clear aperture diameter, 19 cm |
| Rx Optics Loss | >−7 | −7.00 | | Max. tracking sensor (QC) |
| Rx Power | >−125 | −82.58 | dBW | Beam split, 99:1 |
| **SNR Margin** | >10 | 22.21 | dB | - |

**Table 4.** *Cont.*

| (d) COM#1—Communication | | | | |
|---|---|---|---|---|
| Elements | Budget | Available | Unit | Remarks |
| Tx Power | >−3 | 0.00 | dBW | 1 W output |
| Tx Gain | >56 | 92.92 | | Beam divergence, 127.8 μrad |
| Pointing Loss | >−5 | −0.03 | | Pointing error at Table 3 |
| Tx Optics Loss | >−7 | −2.00 | | Front-end, back-end optics |
| Path Loss | <−259 | −258.18 | dB | Max. range, 1000 km |
| Rx Gain | >111 | 111.71 | | Clear aperture diameter, 19 cm |
| Rx Optics Loss | >−7 | −3.50 | | APD |
| Rx Power | >−125 | −59.08 | dBW | Beam split, 99:1 |
| **SNR Margin** | >10 | 15.98 | dB | OOK signal and coded BER $1 \times 10^{-3}$ |

### 3.3. Systems Design Specifications

To ensure the design feasibility of the laser crosslink mission, the system design specifications, listed in Table 5, were evaluated using software simulations, and the systems meet the requirements. The total mass is 11.36 kg, and the dimensions satisfy the 6U CubeSat standard. The operation orbit was selected as the sun synchronous orbit with the local time of ascending node (LTAN) of 18:00 and altitude of 600 km. The laser crosslink can be accomplished with a sufficient SNR margin for all operational inter-satellite distances. At the end of the PAT sequence, the residual LOS jitter becomes smaller than 1 μrad, as achieved by the FSM control.

**Table 5.** Systems requirements and design specifications.

| Parameter | | Requirements | Specifications | Remarks |
|---|---|---|---|---|
| **Physical Properties** | Mass — Payload | <6 kg | 5.70 kg | wet mass |
| | Mass — Bus | <6 kg | 5.66 kg | |
| | Size — Payload | <0.25 × 0.10 × 0.10 m$^3$ | <0.20 × 0.09 × 0.09 m$^3$ | stowed |
| | Size — Bus | 6U standard | <0.25 × 0.12 × 0.34 m$^3$ | |
| **Orbit** | Lifetime | >1 years | 3 years | radiation tolerance |
| | Altitude | 600 ± 25 km | 600 km | sun synchronous |
| | LTAN | 18:00 ± 2 h | 18:00 h | - |
| **Laser Crosslink** | Range | Up to 1000 km | 50~1000 km | - |
| | Capacity | Up to 1 Gbps | 1 Gbps | uncoded BER $1 \times 10^{-3}$ |
| | Residual LOS Jitter (μ, σ) | <(30, 1) [μrad] | <(0.10, 0.96) [μrad] | fine pointing |
| | SNR Margin | >10 dB | >15.98 dB | communication |
| **GNC** | Body Pointing | <75 arcsec (3σ) | <63.5 arcsec (3σ) | slewing, LOS error |
| | Stability | <5 arcsec (1σ) | <2 arcsec (1σ) | slewing, LOS jitter |
| | Relative Navigation | <10 m (3σ) | <0.90 m (3σ) | each axis |
| | Propellant | >5.2 m/s | 6.6 m/s | 10% of residual |
| **Electrical Interface** | Data Communication | 1 Mbps CAN, SPI | 1 Mbps CAN, SPI, UART, RS422, I$^2$C | Ethernet, JTAG for debugging |
| | Power Supply | 3.3 V, 5 V, 12 V | 3.3 V, 5 V, 12 V, Battery Voltage | switchable, latch-up protection |
| **RF Communication** | S-band (TMTC/ISL) | >0.5 Mbps/5 kbps | 1 Mbps/10 kbps | margin > 7.01 dB |
| | X-band (Mission Data) | >100 Mbps | 90–135 Mbps | margin > 5.13 dB |
| | UHF (Redundancy) | >2 kbps | 4.8–9.6 kbps | margin > 9.56 dB |
| **Electrical Power** | Generation (Average) | >16 W | >21.6 W | sun-pointing |
| | Peak Draw | <4 A (protected) | <1.68 A | laser crosslink |
| | | <10 A (unprotected) | <2.72 A | |
| | Depth of Discharge | <20% | <18.5% | 77 Wh battery pack |

The GNC system of the bus assisted the laser crosslink by precisely correcting LOS errors. In particular, the relative navigation system using GPS L1/L2 signals compensates for the effects of ionospheric delay from a long baseline and estimates at submeter levels, mitigating the FOU. The propellant budget for orbit maneuvers has approximately 20% of margin, considering a residual at the end of life. Data communications are conducted mainly through CAN-bus interfaces, and the power supply system includes latch-up protection to prevent over-power draws. RF communication systems consist of S-band, X-band, and UHF radio. From the worst-case analyses, it was confirmed that the link margin of each channel exceeded 5 dB. One of the S-band radios is utilized for relative navigation; then, the data rate is faster than 10 kbps, enabling sharing of the GPS measurement data every 5 s. When they orient to the sun, the power generation is maximized up to 21.9 W at the operation orbit. The selected power system ensures battery capability and lifetime from the depth-of-discharge (DOD) analysis. The peak current draw is lower than the limit for either protected or unprojected channels with any system operation. The preliminary design ensured the performance of the laser crosslink systems, supported by the formation of a nanosatellite bus system. The payload and bus design details are described in the following sections.

## 4. Laser Crosslink Payload

### 4.1. Payload Architecture

The laser crosslink payload has a monostatic architecture that shares communication and PAT beam paths with a single aperture. This design scheme provided closed-loop feedback FSM control without steady-state beam-pointing errors. To operate each PAT sequence, the payload has variable beam-divergence angles. In addition, the SLA of each satellite has a different wavelength, telecom optical C- and L-bands. This approach removes the primary internal reflection effects, which require making a pair of each other for crosslink establishment. Figure 9 shows the payload configurations with allocations for each part. The primary mirror is segmented into three parts, and the secondary mirror is attached to the boom deployment mechanism, thereby saving space during the launch phase. The optical components are arranged on an optical bench. The support plate acts as a mechanical interface with the bus, and the main material is Invar-36 to mitigate on-orbit thermal deformations with structural stiffness.

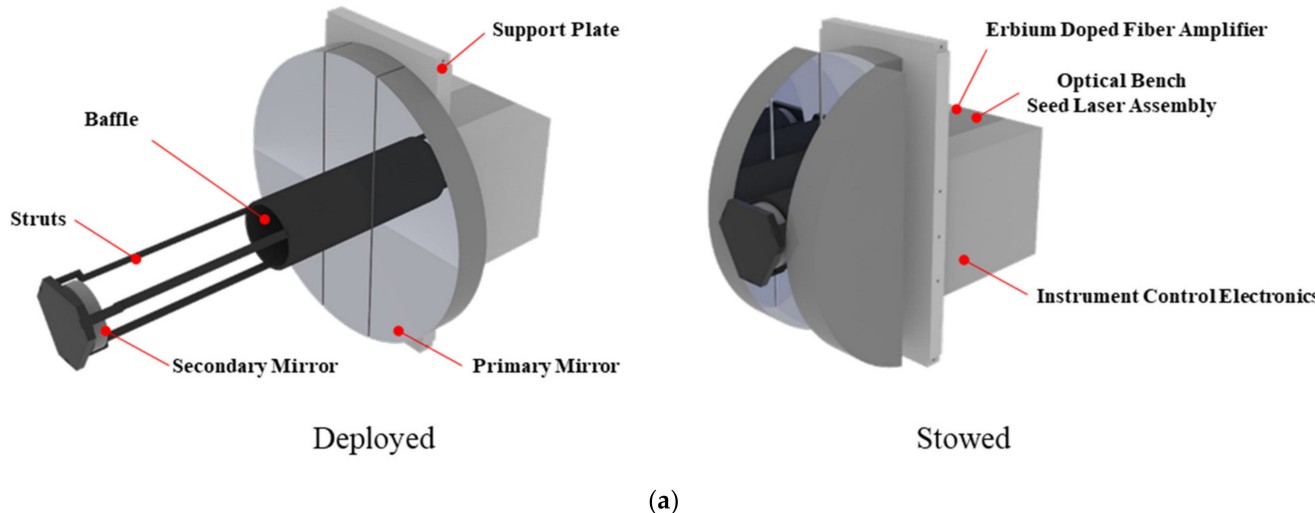

(a)

**Figure 9.** *Cont.*

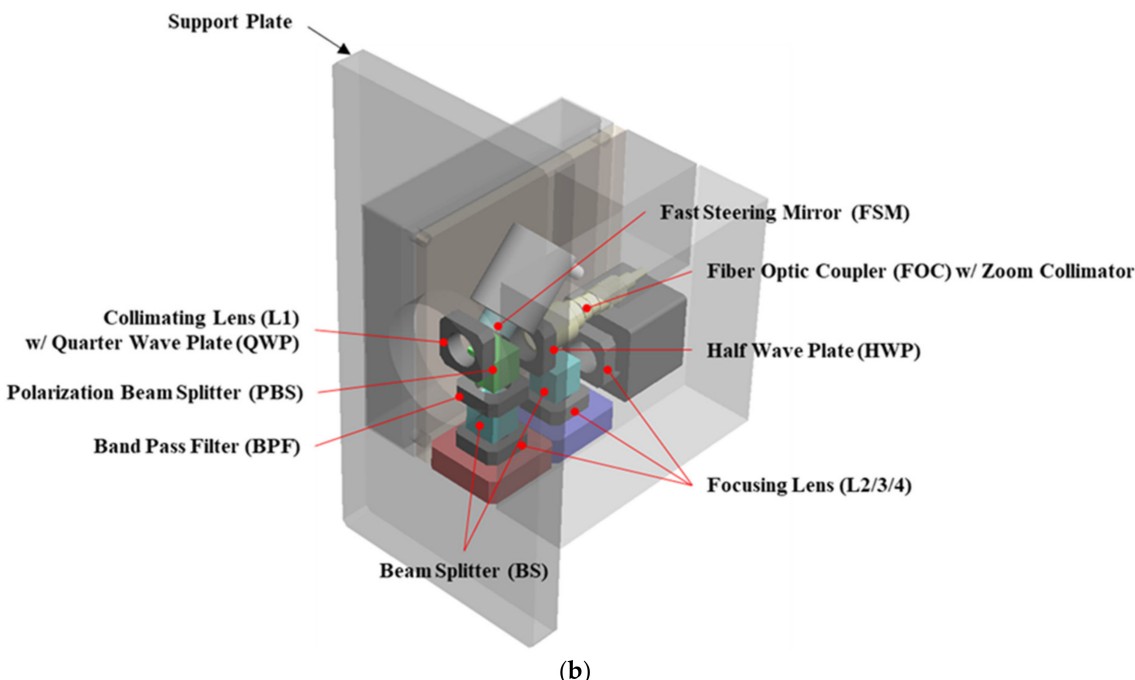

**(b)**

**Figure 9.** Configurations of the laser communication payload: (**a**) stowed and deployed exterior configurations; (**b**) interior configuration of the optical bench.

Table 6 summarizes the payload design specifications. The SLA comprises a laser with 1 mW of optical power; the EDFA amplifies the seed laser up to 1 W. Mass is approximately 5.7 kg and the power draw is less than 39.1 W, meeting the 6U standard constraints. The modulation is basically OOK, which is mostly adopted for laser crosslink systems. From the optical link budget analysis, the design parameters met the system requirements and constraints, including the SNR margin, size, and pointing accuracy. The divergence angle of the laser beam generated by SLA can be adjusted in two ways: coarse PAT (CPAT) and fine PAT (FPAT). The field of view for each detector was designed based on the bus attitude control performance and FSM tip-tilt angles. While the beam spots on each detector are located within the field of regard (FOR) or allowable tip-tilt angles, the FSM can correct them to be close to the center of the detectors.

**Table 6.** Payload design specifications.

| Parameter | | Specifications | Remarks |
|---|---|---|---|
| | **Data Rate** | 1 Gbps | at 1000 km |
| | **Unassisted Pointing** | <±637 µrad | body pointing (3σ) |
| | **Coarse Assisted Pointing** | <±329 µrad | jitter (3σ) |
| | **Pointing Accuracy** | <±1 µrad | residual jitter (1σ) |
| | Size/Mass | <3U/<5.7 kg | (stowed) budget < 6 kg |
| | Power Budget | <39.1 W | budget < 45 W |
| | Electrical Interface | 3.3 V, 5 V, bus VBAT (12.8-16 V) SPI | <2.5 A (latch-up protection) optionally RS-422 |
| | Tx Power | <1 W | 0 dBW |
| **Laser Communication Terminal** | Tx Wavelength | 1550 nm (C-band), 1570 nm (L-band) | |
| | Tx Beam Div. Angle | 8 mrad (CPS) and 128 µrad (FPS) | |
| | PAT Scheme | Hybrid open-/closed-loop | using laser beam |
| | Coarse PAT FOV | <±1745 µrad | focal plane array |
| | Fine PAT FOV | <±873 µrad | quad cell |
| | FSM FOR | <±1047 µrad | FSM spec. and optics design |
| | FSM Resolution | <0.5 µrad | FSM spec. and optics design |
| | Rx Aperture | Φ190 | deployable space telescope |
| | Modulation | On-off keying (OOK) | |

### 4.2. Optomechanical and Electronics Design

To enhance the received optical power, deployment mechanisms were applied to the FEO, which has a large aperture that is difficult to achieve with the conventional 6U CubeSat platform. The clear aperture size was larger than Φ190, and the main material was Zerodur. The distance between the primary and secondary mirrors was approximately 125 mm, given the constraints on the stowed and deployed configurations. Considering the on-orbit thermal environment, CFRP, which is a thermally stable material, was applied to the baffle and secondary mirror deployment mechanism. Figures 10 and 11 show the payload optics design with ray tracing and spot quality analysis. The field stop of the focused beam was located at the front of the back-end optics (BEO) Lens 1. With the tip-tilt control of the FSM within the FOR, spots on the CAM and APD are formed on the focal plane, and the edges of the spots do not exceed the pixel size of the detector. Decentering is required to distinguish the spot position in each section of the QC. When the segmented mirrors were misaligned after deployment, the spots were distorted, as shown in Figure 11c. This distortion degrades the performance of the PAT algorithm by blurring the spots on the detector. The spot quality with DST tolerance is considered in the PAT performance analysis, which determines the spot centers during the PAT sequence.

The electronics handling LCT instruments are integrated in terms of instrument control electronics (ICE), as presented in Figure 12, which includes the optical train. The FPGA manages all electronics, such as the FSM, EDFA, detectors, and SLA. The CPU is utilized as an electrical interface with the nanosatellite bus, including power and thermal management, SPI data communication with the bus, and a deployment mechanism. Because the bus provides an unregulated battery bus voltage, the LCT has a latch-up protection switch. Furthermore, the FPGA executes the PAT algorithm by activating the FSM with an AOA estimation for feedback control.

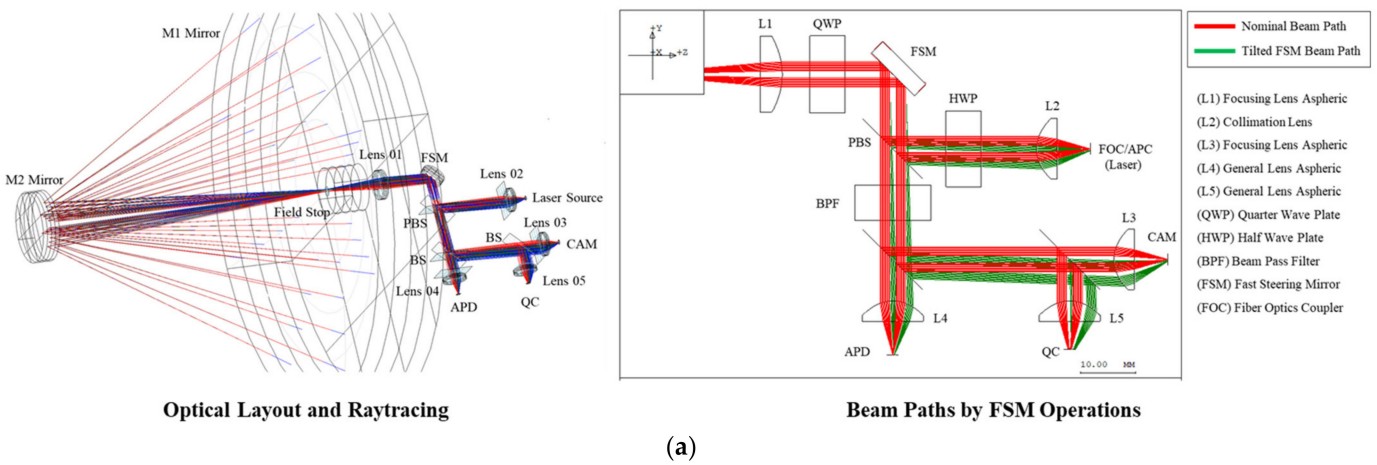

**Optical Layout and Raytracing**　　　　**Beam Paths by FSM Operations**

(**a**)

**Figure 10.** *Cont.*

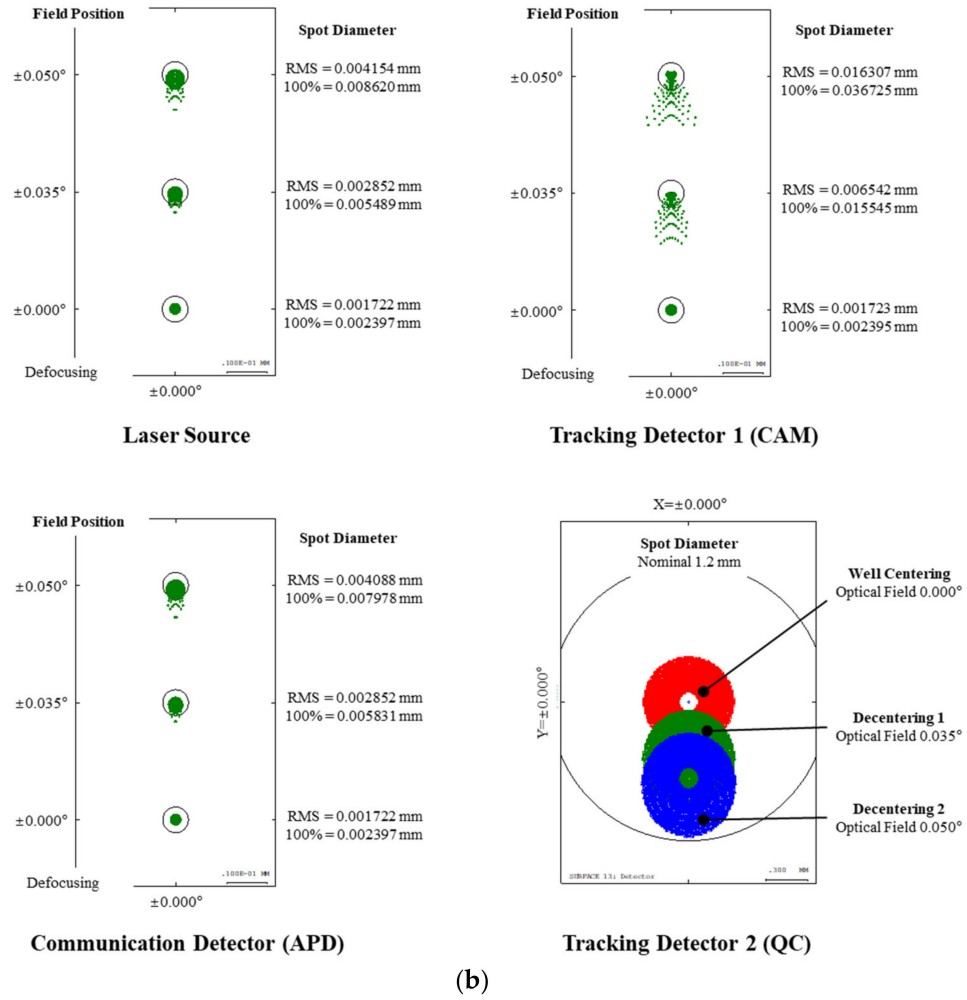

**Figure 10.** Payload optical system design and analysis: (**a**) optical layout and beam paths; (**b**) spot position and quality analysis by defocusing and decentering with a tip-tilt platform (FSM).

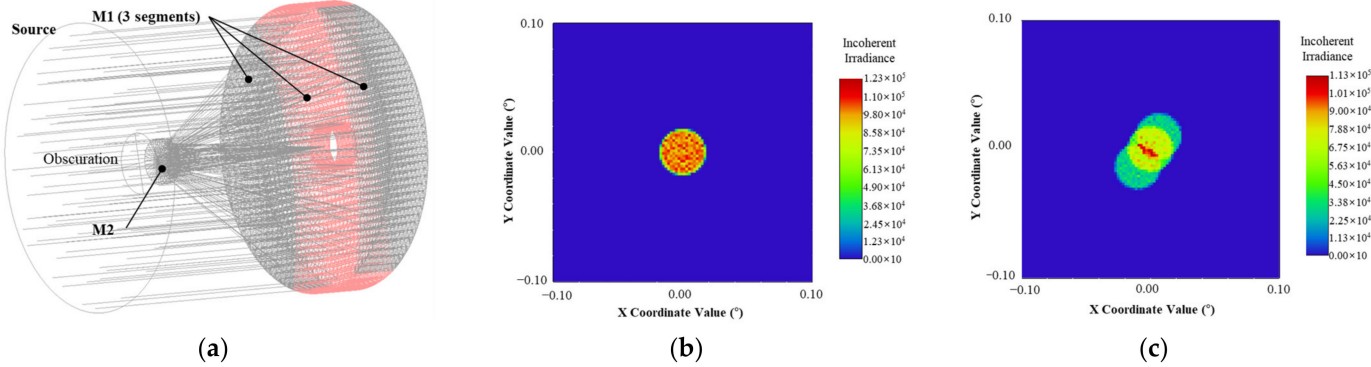

**Figure 11.** Payload optical system design and spot distortion analysis for the deployable space telescope: (**a**) raytracing for the deployable space telescope with three segmented mirrors; (**b**) nominal spot configuration on tracking sensor (CAM); (**c**) distorted spot configuration on tracking sensor (CAM).

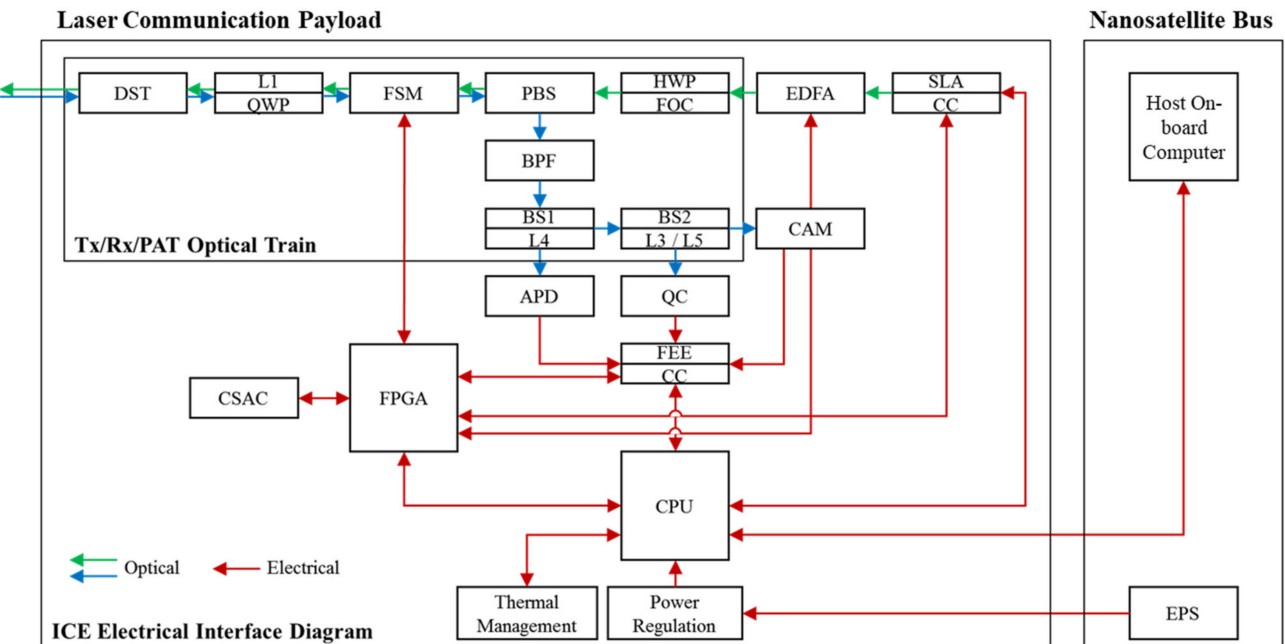

**Figure 12.** Diagram of optical path and electrical interface for the laser communication payload.

### 4.3. PAT Algorithm

For highly precise beam pointing and tracking during laser crosslinking, the LCT estimates the beam spot positions on the detectors and corrects the offset using the FSM. However, micro-vibrations, such as random jitters induced by orbital perturbations and RW control, might increase pointing errors even when a laser crosslink is established. During the FPS, when the QC is available, the FSM is operated to reject jitters at a frequency faster than 200 Hz to avoid resonance with the systems, and the operating frequency is approximately 10 times higher than the natural frequency of the systems. Considering the jitter characteristics of the RWs, the sampling rates of CAM and OC were set to 100 Hz and 2000 Hz, respectively. A PI controller was adopted in the FSM control algorithm to mitigate steady-state errors. Figure 13 shows the power spectral density (PSD) and residual jitters. With FSM operations, the jitter response is significantly reduced below the FSM control frequency domain. Nanosatellite body-pointing error profiles were applied to the FSM control simulations. Finally, the total pointing error during the FSM operations was reduced to less than 1 μrad (0.2 arcsec), as presented in Figure 13c.

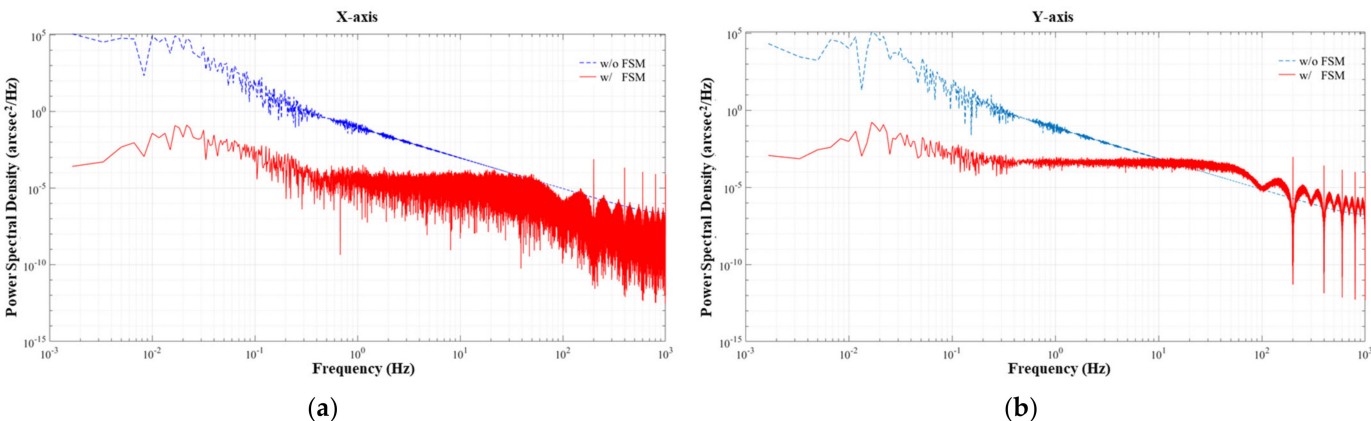

**Figure 13.** *Cont.*

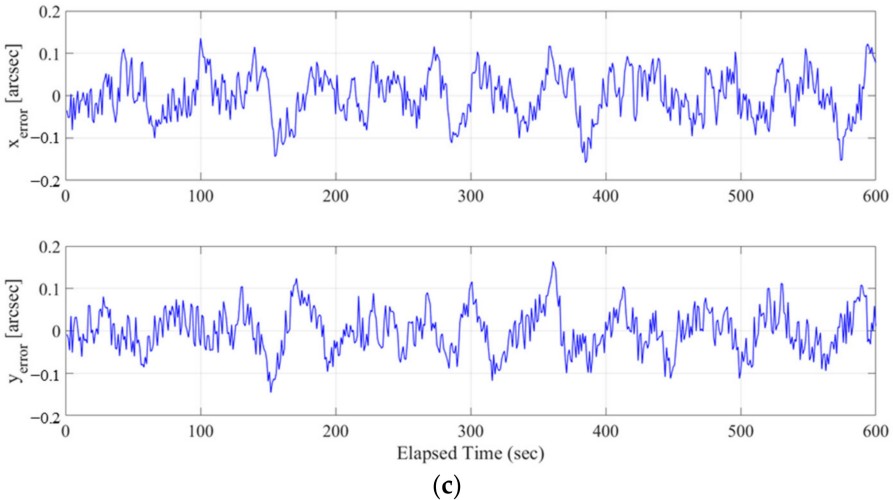

(**c**)

**Figure 13.** Preliminary Matlab/Simulink analysis results for the PSD in frequency domain and errors in time-series according to the FSM tip-tilt control to reject jitters in the fine PAT stage: (**a**) x-axis (roll axis of the bus); (**b**) y-axis (pitch axis of the bus); (**c**) residual jitters in time-series with FSM controls.

## 5. Formation Flying Nanosatellite Bus

### 5.1. Bus Architecture

Figure 14 shows the configurations and body reference coordinates of the nanosatellites. Although both satellites have the same architecture, their star tracker aperture and GNSS antenna are located opposite to each other to obtain visibility over the mission operations. The integrated attitude determination and control subsystem (ADCS) is a single box containing attitude actuators and sensors with its own processor for algorithm execution. Three sun sensors were attached to acquire the sun vector from each state. The designed bus structure provides space for safely mounting the laser crosslink payload, which includes a deployable space telescope. The two deployable solar panels generate electrical power to maintain the battery state of charge (SOC) over 50%, even at the end of the lifetime (EOL). Furthermore, the panels act as baffles by preventing direct sunlight on the payload optics during mission operations.

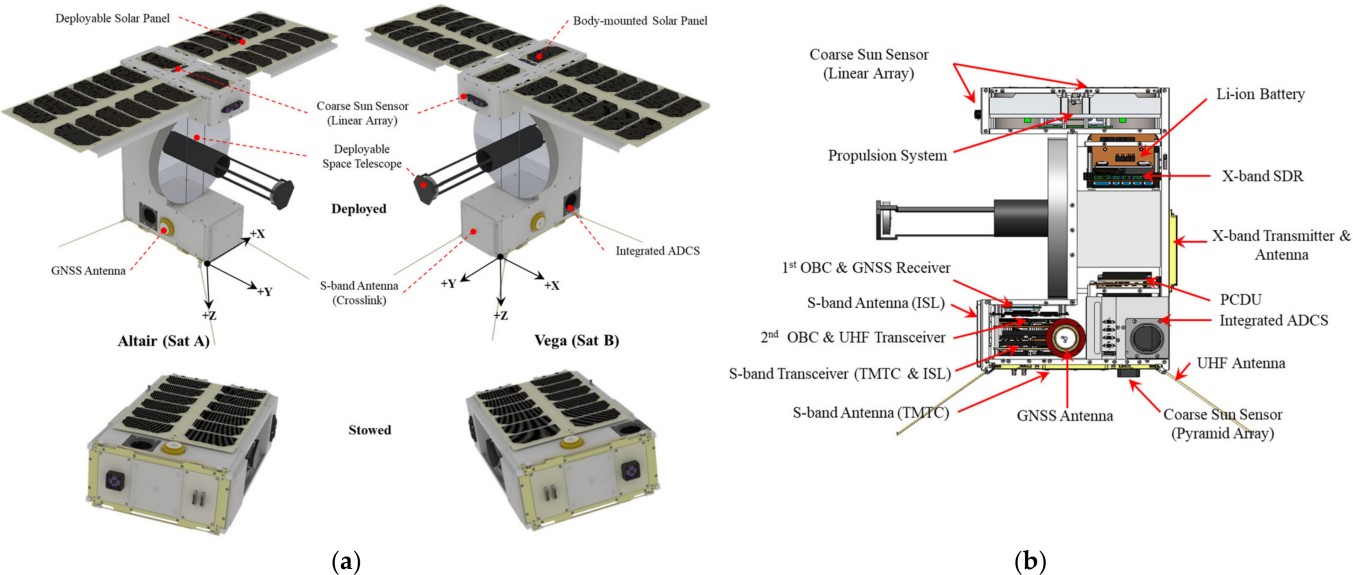

(**a**)　　　　　　　　　　　　　　　　　　　　　　　　　　(**b**)

**Figure 14.** Configurations and body reference frame coordinates of nanosatellites: (**a**) stowed and deployed exterior configurations of the Altair and Vega; (**b**) interior configurations of the Vega. The body

reference coordinates of both satellites are assigned, considering the optical axis (+Y) and antenna boresight (+Z). The GNSS antenna and star tracker aperture heads are opposite each other.

The electrical interfaces for the power supply and data communication are shown in Figure 15. Two deployable solar panels and one body-mounted panel are connected with buck-boost converters on the power conditioning and distribution unit (PCDU) for battery charging. The PCDU manages the power supply for each component with latch-up protection to avoid damage from current or voltage during the mission lifetime. Before the on-orbit operation, dual kill switch mechanisms deactivate the PCDU and battery to prevent battery discharge. By applying two-wire bus interfaces, such as CAN and I²C, the wiring is significantly reduced compared to serial interfaces. To mitigate the susceptibility to bus faults of I²C interfaces, they are applied only to internal or back-up communication interfaces, including a redundancy system [15]. The X-band radio also provides a high-speed communication interface for future applications.

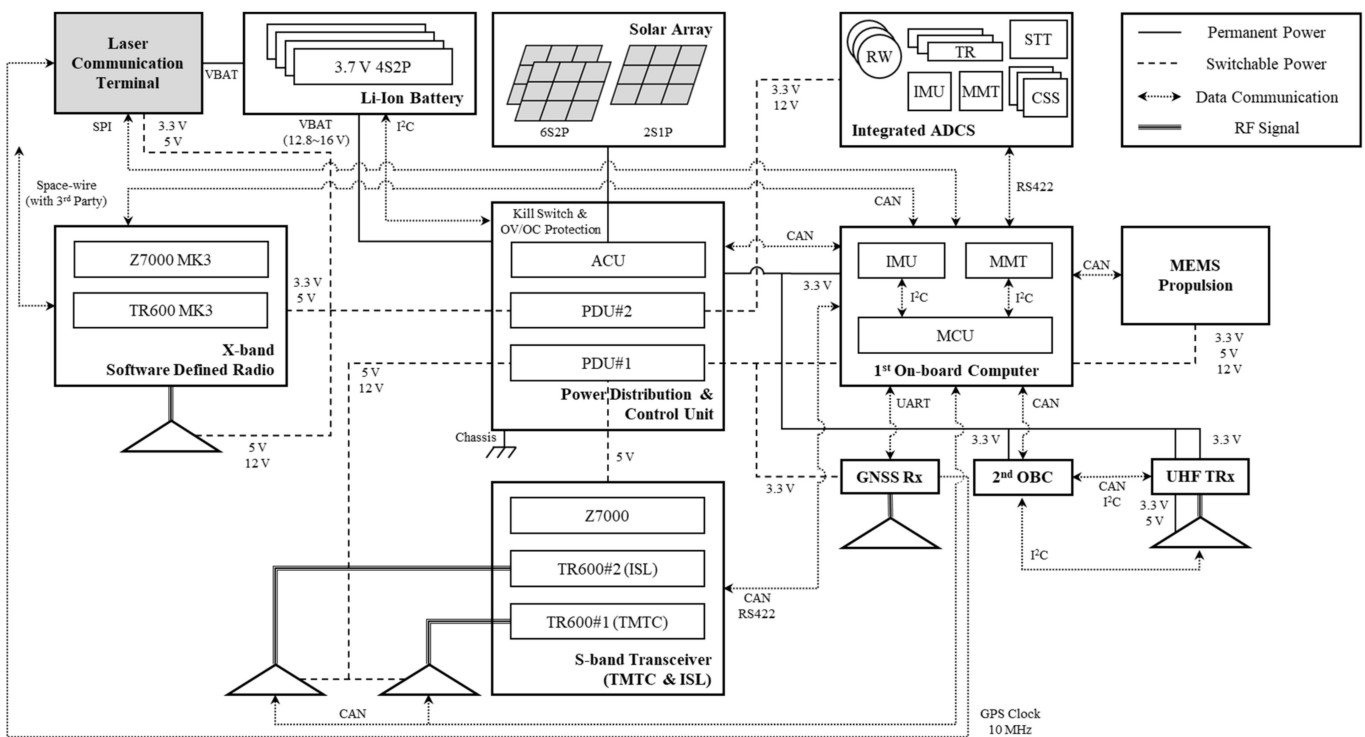

**Figure 15.** Diagram of the electrical interface for the nanosatellite bus. Except the primary OBC and redundancy systems, all power supplies are switchable with latch-up protections. Data communications are mainly implemented by CAN bus.

For the LCT, an unregulated battery bus voltage is provided as the main power and a 3.3 V channel is used to control the device by the primary OBC. Through the SPI interface, the LCT provides an AOA for PAT implementation. Moreover, a GNSS receiver is connected so that the interfaces can synchronize with each other using the GPS clock.

### 5.2. Subsystems Design

As in the aforementioned bus architecture, bus subsystems are designed by applying the COTS products to establish precise formation flying. The details of each bus subsystem are as follows.

### 5.2.1. Guidance, Navigation, and Control Subsystem (GNC)

The GNC subsystem is composed of integrated actuators and sensors for attitude determination and control and a propulsion system for orbit maneuvers. The GNC algorithm for formation flying, similar to relative navigation, is computed using the primary host

OBC. The attitude determination and control for the pointing maneuver are executed by the integrated module called the XACT-50, manufactured by Blue Canyon Technologies (BCT), which ensures the most precise pointing performance on the nanosatellite platform.

Figure 16 shows a diagram of the formation flying architecture of the VISION system. Three coarse sun sensor (CSS) arrays are attached to acquire the sun vector. Arcsecond-level attitude determination can be achieved using a star tracker and MEMS gyro. While conducting the laser crosslink, the LCT provides the AOA to the host OBC every 10 Hz and the bus corrects the LOS error every 1 Hz. The 3-axis RWs are balanced and provide high momentum and torque capacities, having a low jitter characteristic with viscoelastic dampers [16]. As shown in Figure 17a, the FOU includes the region of the LOS errors yielded by the relative navigation, body pointing, and residual of FSM control. The body pointing, given the relative navigation and hardware performances, is evaluated as presented in Figure 17b,c. The body-pointing errors are smaller than 75 arcsec during the PAT sequence, and the beam spot can remain within the active area of the tracking sensors.

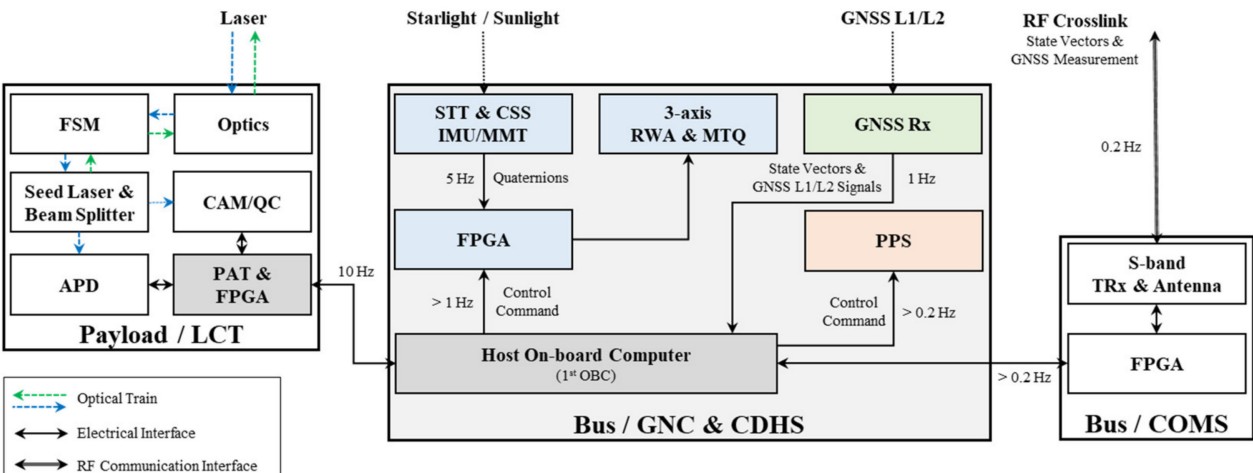

**Figure 16.** Diagram of the GNC architecture for formation flying. The host OBC commands the integrated attitude determination and control module for body pointing. During the PAT sequence, the LOS vector is yielded from the AOA and relative navigation, and the body pointing is executed faster than 1 Hz.

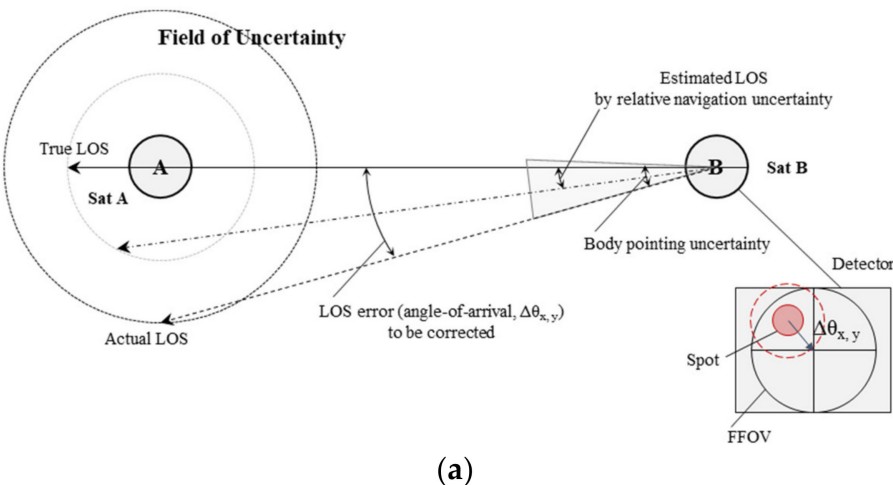

(**a**)

**Figure 17.** *Cont.*

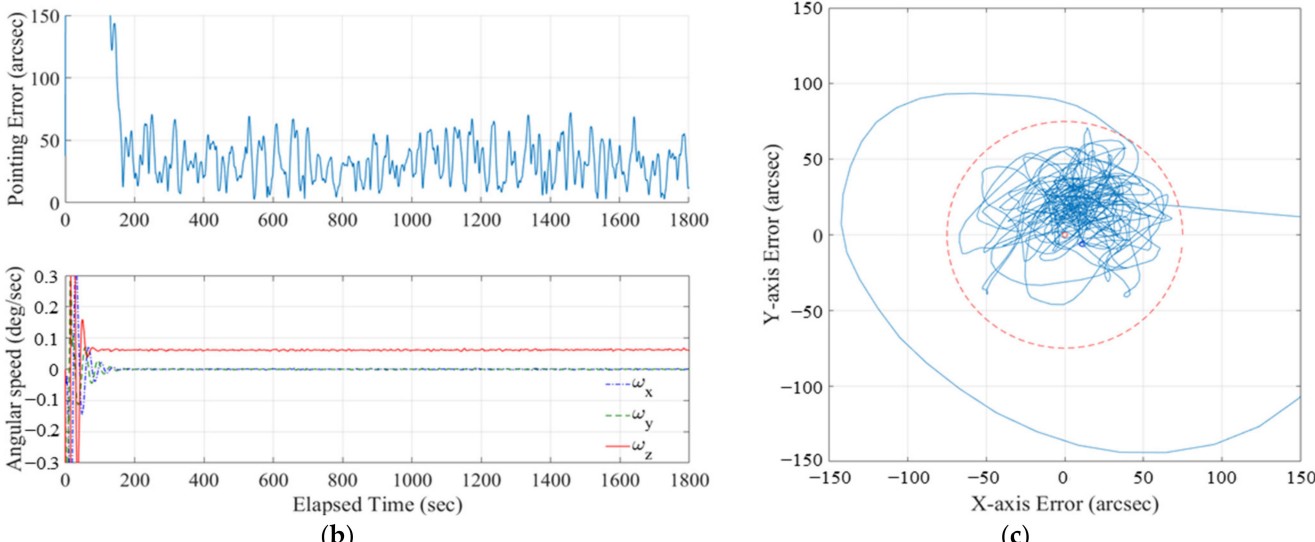

**Figure 17.** Diagrams and simulation results of the line-of-sight (LOS) errors for the PAT sequences: (**a**) LOS errors with uncertainties of relative navigation and body pointing can be corrected using angle-of-arrival (AOA) on tracking sensor (CAM); (**b**) body-pointing errors and angular velocity profiles in time-series; (**c**) body-pointing errors on a projected plane (body coordinated x-y).

The S-band transceiver and antenna are utilized to transfer the GPS L1 and L2 signals obtained by the GNSS receiver. The differential ionosphere makes usually negligible effects for a short baseline of a few kilometers. However, it is dependent on ionospheric conditions, and an ionospheric uncertainty can be corrected with a dual-frequency GPS receiver for a long baseline, which is greater than a few kilometers [17]. In addition, for precise relative position estimations, the algorithm based on DGPS corrects the delays induced by data acquisition, parsing, and RF crosslinks. Table 7 summarizes the results for each orbit scenario. Relative navigation achieves submeter-level estimation performance.

**Table 7.** Simulation results of the relative navigation by inter-satellite distances.

| Range (km) | State | Relative Navigation Error (Mean, $3\sigma$) | | |
|---|---|---|---|---|
| | | **Radial** | **In-Track** | **Cross-Track** |
| **50** | Position [cm] | $0.62 \pm 11.50$ | $0.03 \pm 5.28$ | $0.06 \pm 11.59$ |
| | Velocity [cm/s] | $0.24 \pm 0.84$ | $-0.00 \pm 0.39$ | $0.00 \pm 0.58$ |
| **100** | Position [cm] | $0.85 \pm 10.91$ | $-0.00 \pm 5.57$ | $0.03 \pm 15.64$ |
| | Velocity [cm/s] | $0.48 \pm 0.80$ | $-0.01 \pm 0.41$ | $0.00 \pm 0.59$ |
| **200** | Position [cm] | $2.24 \pm 12.70$ | $0.01 \pm 5.81$ | $0.04 \pm 25.09$ |
| | Velocity [cm/s] | $0.95 \pm 0.94$ | $-0.01 \pm 0.43$ | $0.00 \pm 0.64$ |
| **500** | Position [cm] | $5.06 \pm 15.82$ | $-0.06 \pm 5.80$ | $0.69 \pm 49.47$ |
| | Velocity [cm/s] | $2.39 \pm 1.22$ | $-0.03 \pm 0.45$ | $0.01 \pm 0.63$ |
| **1000** | Position [cm] | $10.53 \pm 24.57$ | $0.06 \pm 6.30$ | $0.35 \pm 89.34$ |
| | Velocity [cm/s] | $4.79 \pm 1.94$ | $-0.05 \pm 0.53$ | $0.02 \pm 0.61$ |

In addition, the propulsion system is utilized for orbit maneuvers. The propulsion system has four MEMS nozzles that provide a maximum thrust of 1 mN along the z-axis for each nozzle. It is a cold-gas type which can vaporize the n-butane by heating the titanium-based tank and nozzles. In addition, continuous thrust is enabled with a pulse width of 10 ms. As presented in Table 8, the total accumulated propellant over the orbit scenario is approximately 5.13 m/s, which is less than the available propellant budget. The remaining propellant is sufficient for reentry maneuvers, following the "25-year rule".

**Table 8.** Propellant budget for each orbit maneuver scenario.

| Scenario | Budget (m/s) | ΔV (m/s) | Attempt (Times) | Total ΔV (m/s) | Margin (%) |
|---|---|---|---|---|---|
| Drift Recovery | 3.0 | 2.55 | 1 | 2.55 | 15.0 |
| Station Keeping [1] | 0.5 | 0.38 | 1 | 0.38 | 24.0 |
| Reconfiguration | 2.5 | 0.11 | 20 | 2.20 | 12.0 |
| Residual and Disposal | 0.6 | | | | 100.0 |
| **Accumulated ΔV (m/s)** | 6.6 | | | 5.13 | 22.3 |

[1] This stage includes the station-keeping maneuvers for maintaining inter-satellite distance.

### 5.2.2. Electrical Power Subsystem (EPS)

The satellites have two deployable and one body-mounted solar panels integrated with highly efficient multiple-junction GaAs cells. They are connected to buck-boost regulators, which guarantee a DC-to-DC conversion efficiency of 90%. The orbit average power generation was up to 21.9 W. The battery pack selected was 4S-2P lithium-ion cells with a capacity of 77 Wh capacity with 14.8 V as a nominal voltage. Table 9 summarizes the electrical power budget analyses for the operation scenarios. For the analysis, the maximum eclipse duration was applied over the mission's lifetime. With duty-cycled operation, the average power consumption was calculated. Given the DC-to-DC conversion efficiency of the regulators, the DOD for each mode was smaller than 20% according to the system requirements.

**Table 9.** Electrical power budget for each operation mode.

| Parameter | Operation Scenarios | | | | | |
|---|---|---|---|---|---|---|
| | LEOP [3] | Maneuver | Standby | Mission | Comm. | Safe |
| **Power Generation (W)** | 5.81 | 17.31 | 21.64 | 17.31 | 17.31 | 21.64 |
| **Power Consumption (W)** | 3.56 | 9.29 | 5.81 | 16.24 | 9.70 | 4.47 |
| **Discharge (Wh)** | −7.18 | −16.64 | −11.71 | −26.67 | −17.37 | −9.01 |
| **Charge (Wh) [1]** | 3.14 | 13.17 | 19.76 | 13.17 | 13.17 | 19.76 |
| **Margin (Wh)** | −4.03 | −3.47 | 8.05 | −13.50 | −4.20 | 10.75 |
| **Depth of Discharge (%) [2]** | 5.51 | 4.75 | | 18.45 | 5.74 | |

[1] The actual charge values are estimated at end-of-lifetime with conversion efficiency of buck-boost converters and man-made performance degradations. [2] The battery pack capacity is 77 Wh. [3] The orbit scenario is assumed to detumbling periods after separation.

### 5.2.3. Communication Subsystem (COMS)

Transceivers for each band are based on software-defined radio (SDR), which simply changes its RF features in orbit. Figure 18 shows the frequency, data rate, and modulations for each RF application. Most communications are established in the UHF and S-bands. In particular, the S-band transceiver includes two modems in one unit for either the telemetry and telecommand (TMTC) or the crosslink (inter-satellite link, ISL), saving internal space and power consumption. UHF communication was adopted for early orbit operations and back-up communication for contingencies. Finally, X-band communication is available for future applications but is not currently utilized. The RF link budget analysis was conducted to ensure link availability in orbit, as summarized in Table 10. For both UHF- and S-band communications, the link budget should be higher than 6 dB; for the X-band, it should be higher than 4 dB. By applying the specifications of each device, the link budget meets the requirements of data rate and modulations; an S-band crosslink would be available in the range of 1000 km.

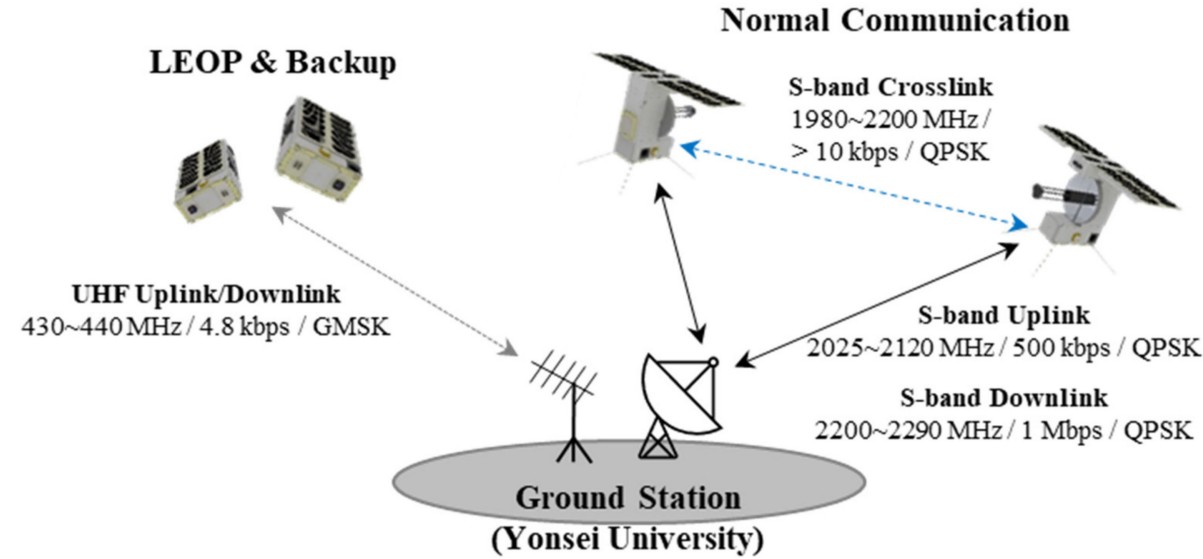

**Figure 18.** Configurations of the RF communication subsystem.

**Table 10.** RF communication link budget for each communication module.

| Elements | Unit | Downlink [1] | | | Uplink [1] | | Crosslink [2] |
|---|---|---|---|---|---|---|---|
| | | **UHF** | **S-Band** | **X-Band** | **UHF** | **S-Band** | **S-Band** |
| Modulation | - | GMSK | QPSK | 8-PSK | GMSK | QPSK | QPSK |
| Frequency | MHz | 437.0 | 2200.0 | 8250.0 | 437.0 | 2100.0 | 2200.0 |
| Data Rate | Kbps | 4.8 | 1000.0 | 1,000,000.0 | 4.8 | 500.0 | 10.0 |
| Tx Power | W | 1.0 | 1.0 | 2.0 | 27.0 | 27.0 | 1.0 |
| Tx Gain | dBi | 0.0 | 8.0 | 13.0 | 18.9 | 36.0 | 8.0 |
| EIRP | dBm | 29.5 | 37.8 | 41.7 | 57.8 | 76.3 | 7.4 |
| Path Loss | dB | −153.2 | −163.1 | −169.5 | −153.2 | −162.7 | −160.1 |
| Rx Gain | dB | 18.9 | 36.0 | 51.0 | 0.0 | 8.0 | 8.0 |
| Eb/N0 | dB | 17.3 | 14.8 | 14.0 | 41.0 | 37.6 | 18.6 |
| **Link Margin** | dB | 9.6 | 7.0 | 4.2 | 33.2 | 29.8 | 6.7 |

[1] Minimum elevation for the downlink and uplink is set to 20 degrees. [2] Maximum range for the crosslink is set to 1100 km.

### 5.2.4. Command and Data-Handling Subsystem (CDHS)

The OBC has the following capabilities and features: low power consumption within 0.5 W, 400 MHz clock speed with the ARM cortex A5, embedded RT-patched Linux OS, and docking for the GNSS receiver, which supports multiple channels for parallel interfaces such as CAN-bus and I$^2$C. The FSW is based on the core flight system (cFS) developed by NASA to be used as the main platform for the FSW. Thus, the FSW has a simplified architecture and is robust, providing multitasking, such as the computation of the formation flying GNC algorithms. With the basic functions in the cFS, the software bus (SB) provides an interface between each module, enhancing the robustness of the FSW and reducing the development cost. In addition, the back-up OBC integrated with the UHF transceiver was adopted to handle on-orbit contingencies, acting as a hardware watchdog timer. The FSW architecture and the configurations of the two computers are shown in Figure 19.

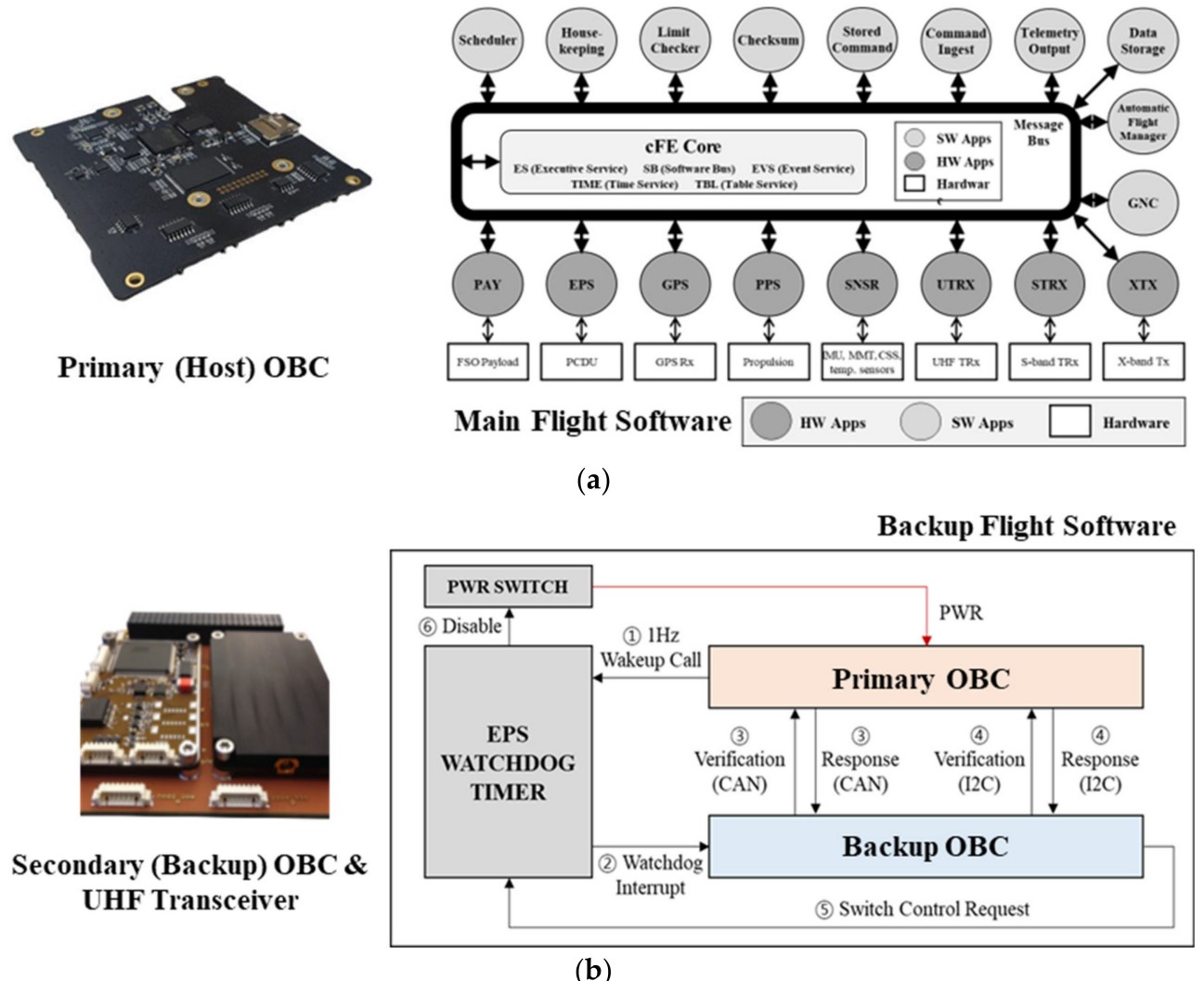

**Figure 19.** Diagrams and configurations of the flight software architecture and on-board computer: (**a**) the primary on-board computer and the cFS-based flight software architecture. Each application is managed on the software message bus; (**b**) the back-up on-board computer and the flight software handover procedure. The primary OBC sends a wakeup to the EPS watchdog timer every 1 sec, and then, the response interrupts the backup OBC. Through CAN and I2C communications, two OBCs check status each other, and the recovery and isolation are conducted by handling power switches.

5.2.5. Structure and Mechanism Subsystem (SMS)

The structural parts, including the frame and hinge mechanisms, are made of aluminum 6061 alloy. The surfaces of these parts are anodized to prevent cold welding between the CubeSat deployers. Considering the payload integration, the frame design has a skeleton configuration with a high degree of freedom during the assembly process. Given the internal space, as depicted in Figure 14b, the avionics are assembled by functions; for instance, the stacked boards for CDHS and COMS are located on the +Y-axis, and the integrated ADCS module was adopted. By conducting a launch environment simulation with NX10.0 NASTRAN, the first mode frequency ($f_0$) with the stowed configuration during the launch phase was analyzed above 80 Hz, above the recommended value to avoid resonance with a launch vehicle.

### 5.2.6. Thermal Control Subsystem (TCS)

Passive thermal control was applied to each satellite using an anodized aluminum frame and a black-colored FR-4 PCB. The battery board includes heaters for heat dissipation, which maintain the temperature of the battery cells above 0 °C. The on-orbit thermal transient simulation was conducted using an NX10.0 Space Thermal System. Figure 20 shows the exterior temperature contours for the hottest and coldest cases with seasonal eclipse variations. The temperature ranges summarized in Table 11 are within the operating temperature range, which have thermal margins above 10 °C, whereas the deactivated components are within the survival temperature.

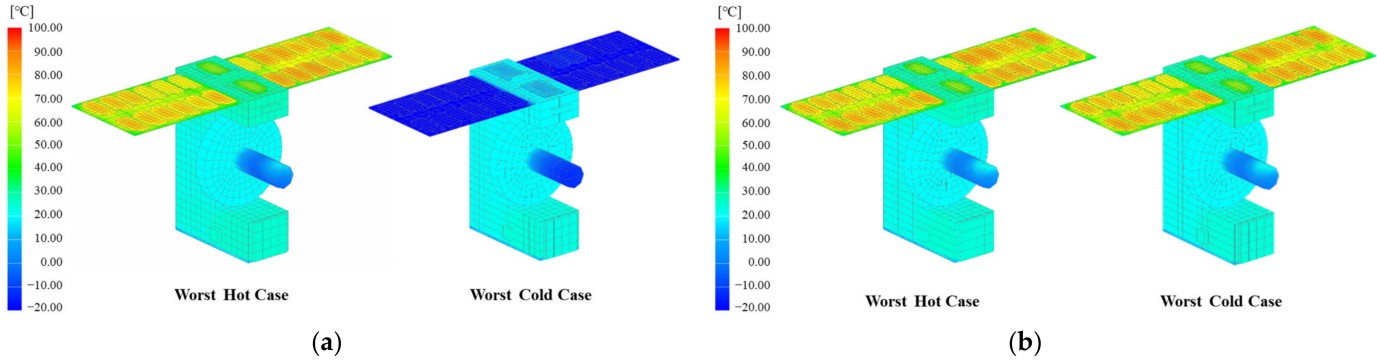

**Figure 20.** Exterior temperature contours of on-orbit thermal analysis for worst hottest and coldest case. The worst hottest case is for maximum solar flux and internal heat loads. The worst coldest case is for minimum solar flux and internal heat loads: (**a**) solar irradiance conditions for the summer season which have eclipse periods; (**b**) solar irradiance conditions without eclipse periods.

**Table 11.** Summary of thermal analysis results for worst hottest and coldest cases.

| Subsystem | | Operating Temperature (°C) | | Worst Case Analysis (°C) | |
|---|---|---|---|---|---|
| | | **Min.** | **Max.** | **Min.** | **Max.** |
| **Payload** | Laser Communication Terminal | −40 | +85 | 20.37 | 23.71 |
| **GNC** | Integrated ADCS | −20 | +50 | 21.32 | 22.61 |
| | GNSS Antenna | −40 | +85 | 23.41 | 25.00 |
| | Propulsion | 0 | +50 | 20.33 | 25.40 |
| **CDHS** | Primary OBC and GNSS Receiver | −40 | +85 | 24.28 | 25.70 |
| | Secondary OBC and UHF Transceiver | −30 | +85 | 23.81 | 25.30 |
| **COMS** | S-band SDR | −40 | +85 | 23.87 | 25.48 |
| | S-band Antenna (TMTC) | −40 | +85 | 11.53 | 16.09 |
| | S-band Antenna (Crosslink) | −40 | +85 | 23.21 | 24.85 |
| | X-band SDR | −40 | +85 | 20.50 | 21.48 |
| | X-band Transmitter and Antenna | −40 | +57 | 20.38 | 22.42 |
| | UHF Antenna | −40 | +85 | 12.68 | 17.50 |
| **EPS** | Solar Panel | −40 | +105 | −27.83 | 83.08 |
| | PCDU | −35 | +85 | 21.35 | 22.40 |
| | Battery | 0 | +45 | 20.37 | 23.74 |

### 6. Conclusions

This study proposed design schemes for novel laser crosslink systems with a 6U nanosatellite platform, including formation flying mission scenarios and system design specifications. The aim of the VISION mission is to establish a miniaturized laser crosslink with a 1 Gbps level of super-high-speed data transfer at an inter-satellite distance of 1000 km. Additionally, several space technologies, such as deployable space telescopes, were proposed for future applications. The laser crosslink mission scenarios were presented in

detail from link access to maintenance. An optical link budget analysis was conducted to evaluate system performance. The main contribution of this study is the advancement of spaceborne laser communication systems. The nanosatellites in formation flying and laser crosslink payloads were designed to meet the system requirements under practical limitations utilizing commercial off-the shelf products, which would reduce the cost and effort of system performance evaluation and on-ground verification. In addition, fundamental technologies for space optics were proposed for the sensing of remote areas with super-high resolution. Moreover, owing to their precise formation flying technologies, including orbit maneuvering capabilities, the proposed nanosatellite systems can be utilized as platforms for mega-constellation applications.

These preliminary systems are under development in accordance with the Engineering and Qualification Model (EQM) philosophy. A prototype of the payload was developed. The prototype is expected to demonstrate the on-orbit performance of laser crosslink and PAT using a far-field hardware testbed which can emulate disturbances such as pointing errors and jitter. In addition, the electrical testbed (ETB) of the bus was constructed to test the electrical interfaces among components. An end-to-end (ETE) testing with the FSW is being planned for this year.

**Author Contributions:** Conceptualization, G.-N.K., S.S., J.-Y.C., S.-K.H. and S.-Y.P.; methodology, G.-N.K., S.S., J.-Y.C. and S.-K.H.; software, G.-N.K., J.-Y.C., Y.-E.K., S.C., J.L., S.L. and H.-G.R.; validation, G.-N.K., J.-Y.C., S.K., Y.-E.K., S.C., J.L. and H.-G.R.; formal analysis, G.-N.K., J.-Y.C., S.K., Y.-E.K., S.C. and J.L.; investigation, G.-N.K., S.S. and J.-Y.C.; resources: G.-N.K., S.S., J.-Y.C., S-K.H. and S.-Y.P.; data curation, G.-N.K., S.S. and J.-Y.C.; writing—original draft preparation, G.-N.K.; writing—review and editing, G.-N.K., S.S., J.-Y.C. and S.-Y.P.; visualization, G.-N.K., S.S. and S.K.; supervision, S.-Y.P.; project administration, S.-Y.P.; funding acquisition, S.-Y.P. All authors have read and agreed to the published version of the manuscript.

**Funding:** This research was supported by the Challengeable Future Defense Technology Research and Development Program (912908601) of the Agency for Defense Development, 2020.

**Institutional Review Board Statement:** Not applicable.

**Informed Consent Statement:** Not applicable.

**Data Availability Statement:** Not applicable.

**Conflicts of Interest:** The authors declare no conflict of interest. The funding sponsors had no role in the design of the study; in the collection, analyses, or interpretation of data; in the writing of the manuscript; or in the decision to publish the results.

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
