# Peer review of "Design of Novel Laser Crosslink Systems Using Nanosatellites in Formation Flying: The VISION"

_aerospace, doi:10.3390/aerospace9080423_

Round 1
Reviewer 1 Report
Review of Geuk-Nam Kim, Sang-Young Park, Sehyun Seong et al. "Design of Novel Laser Crosslink Systems Using Nanosatellites 2 in Formation Flying: The VISION"
The article consider the prototype of the VISION inter-satellite communication laser system, which is apparently supposed to be tested in the near future. The features of the system are as follows. It should provide communication at a speed of 1 Gbit / s at distances between satellites from 50 to 1000 km. Satellites are manufactured identical in 6U CubeSat format. The same optical system is used for transmitting and receiving information, and for the pointing system.
To increase the aperture, the optical system is made folding. This idea is not original, a similar scheme is used in the JWST space telescope. It should be noted that it was not offered for CubeSat.
The development has apparently been brought to the level of an engineering computer model, which allows the authors to carry out numerical simulations of various processes in these spacecraft. The results of such modeling are presented in the article in the form of a series of tables, graphs and figures.
Very important for the implementation of the VISION system is the accuracy of mutual aiming of the optical systems of vehicles. Line-of-sight pointing accuracy is claimed to be 30 µrad (6"), and jitter is 1 µrad (0.2"). The authors claim that these values are achieved according to their simulations and provide confirmation of this in tables 3, 4 and in the GNC section of table 5. But it is impossible to check or at least evaluate the values given in these tables - the article does not contain the data necessary for this. What is the size of the registered spot in the focal plane? What Quadrant Cell photoreceivers are you using? How many quanta are registered?
I believe the lack of the possibility of verifying the data presented by the authors is a very serious shortcoming of the article, which requires its revision.
Note also that Figure 17 shows the results of modeling the mutual guidance of the satellites, where the accuracies are on the order of 50", one hundred to about 100 times greater than the claimed 1 µrad.
In addition, there are several obvious blunders in the article.
1. Line 305 says "When the beam is propagated in free space, the signal power is exponentially reduced". In a vacuum, in the absence of absorption, the beam attenuates inversely proportional to the square of the distance.
2. Lines 342-343 say "... navigation system using GPS L1/L2 signals compensates for the effects of ionospheric delay...". For the considered satellites, there is no ionospheric delay of GPS signals, since they fly at an altitude of 600 km, i.e. above the ionosphere.
3. Line 464 says "(solar) panels act as baffles by preventing direct sunlight on the payload optics during mission operations". Since the satellites must face each other with their optical systems, and the radio communication antennas should be directed to the Earth, the solar panels can play the role of blends simultaneously for both satellites in extremely rare cases, with a certain orientation relative to the Sun.
The article also contains technical inaccuracies:
4. The abbreviation "RW" is not described.
5. The decoding of the abbreviation "LTAN" is not given.
Reviewer 2 Report
Figure 1 needs better explanation; do the right hand side images represent the type of data that is transferred by the proposed system? If so, what type(s)?
Line 133: "top-level mission requirement" needs to be followed by "(TMR)" for clarity in Table 1.
After the 1yr life time, how long does it take for the satellites' reentry to happen? It should be good to mention the "25-year rule" or include the orbit decay analysis results?
Line 540: more details for thrusters could be included. Are they MEMS electrosprays? Are they outsourced or built in-house?
Ref [2] Initials missing: Long, M.J.
